# Cheap and Deterministic Inference for Deep State-Space Models of Interacting Dynamical Systems

**Andreas Look**        *andreas.look@bosch.com*
*Bosch Center for Artificial Intelligence*

**Melih Kandemir**        *kandemir@imada.sdu.dk*
*University of Southern Denmark*

**Barbara Rakitsch**        *barbara.rakitsch@bosch.com*
*Bosch Center for Artificial Intelligence*

**Jan Peters**        *peters@ias.informatik.tu-darmstadt.de*
*Technical University Darmstadt*

*Reviewed on OpenReview:* *https://openreview.net/forum?id=dqgdBy4Uv5*

## Abstract

Graph neural networks are often used to model interacting dynamical systems since they gracefully scale to systems with a varying and high number of agents. While there has been much progress made for deterministic interacting systems, modeling is much more challenging for stochastic systems in which one is interested in obtaining a predictive distribution over future trajectories. Existing methods are either computationally slow since they rely on Monte Carlo sampling or make simplifying assumptions such that the predictive distribution is unimodal. In this work, we present a deep state-space model which employs graph neural networks in order to model the underlying interacting dynamical system. The predictive distribution is multimodal and has the form of a Gaussian mixture model, where the moments of the Gaussian components can be computed via deterministic moment matching rules. Our moment matching scheme can be exploited for sample-free inference, leading to more efficient and stable training compared to Monte Carlo alternatives. Furthermore, we propose structured approximations to the covariance matrices of the Gaussian components in order to scale up to systems with many agents. We benchmark our novel framework on two challenging autonomous driving datasets. Both confirm the benefits of our method compared to state-of-the-art methods. We further demonstrate the usefulness of our individual contributions in a carefully designed ablation study and provide a detailed runtime analysis of our proposed covariance approximations. Finally, we empirically demonstrate the generalization ability of our method by evaluating its performance on unseen scenarios.

## 1 Introduction

Many dynamical systems, such as traffic flow (Li et al., 2018; Yu et al., 2018), fluid dynamics (Ummenhofer et al., 2019) or human motion (Jain et al., 2016), involve interactions between agents. *Graph Neural Networks* (GNN) (Battaglia et al., 2018) have recently emerged as a powerful tool in these settings, since they allow learning the dynamics of interacting systems from data only. Recent research has made great advances for modeling deterministic complex systems by being able to extrapolate from systems with a small number of agents and short time horizons to systems with a high number of agents and long time horizons. These methods have been successfully applied to a number of physical systems covering fluids, rigid solids and deformable materials (Sanchez-Gonzalez et al., 2020). However, for many real-world applications, predicting a single future trajectory for each agent is not enough, since the stochasticity in the dynamical system

has significant consequences. For instance, in autonomous driving, the driver's intention (e.g. overtaking, turning, lane changing) is a hidden factor that may induce different modes of driving trajectories.

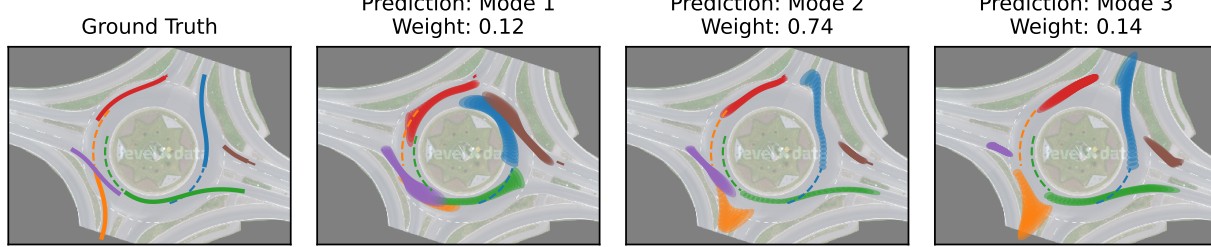

Figure 1: We approximate the predictive distribution of a latent *Graph Deep State-Space Model* (GDSSM) as a Gaussian mixture distribution via deterministic moment matching rules. Given historical information in the form of observed trajectories (dashed lines), our proposed GDSSM architecture predicts the future dynamics while taking interactions between traffic participants into account. We present the true future trajectories (left-most plot) as solid lines. For each mode and traffic participant, we show the predicted 95% confidence interval of our model (three right-most plots). Our model accounts for interactions, for example, the brown vehicle is only entering the roundabout if the blue vehicle is staying in the roundabout (Mode 1). If the blue vehicle is leaving the roundabout, the entering lane for the brown vehicle is blocked by the blue vehicle (Mode 2, Mode 3) and the brown car has to wait. The ground truth data and the map come from the rounD dataset (Krajewski et al., 2020).

In the deep learning literature, multiple architectures have been successfully applied to time-series data with two prominent classes being a) recurrent methods that apply a fixed transition model repeatedly at each time step (e.g. Hochreiter (1991), Chung et al. (2014)), b) history-based methods that aggregate information from the past using either convolutional filters (e.g. Bai et al. (2018), Oord et al. (2016)) or attention modules (e.g. Vaswani et al. (2017), Li et al. (2019)). Recurrent methods capture the internal state of the system at each time point $t$ in a latent state $x_t$. Predicting the output in this manner respects the causal order of the dynamical system, i.e. the latent state $x_t$ of time point $t$ is needed in order to compute the latent state $x_{t+1}$ at the next time point $t+1$. State-space models (e.g. Särkkä (2013)) belong to the class of recurrent methods. They are defined by two probability distributions: the transition model, $p(x_{t+1}|x_t)$, that propagates the latent state forward in time and the emission model, $p(y_t|x_t)$, that maps the latent state into the observational space. Obtaining multistep-ahead predictions for state-space models requires propagating distributions in the latent space. For general transition models, this operation cannot be done in closed-form and existing methods either use numerical integration schemes (e.g. Solin et al. (2021), Look et al. (2022)) or Monte Carlo methods (e.g. Krishnan et al. (2017), Bayer et al. (2021)). For interacting systems, the latent space grows linearly with the number of agents, requiring a high number of MC samples, which can make these methods prohibitively slow. To the best of our knowledge, numerical integration schemes have not been explored for multi-agent dynamical systems. In contrast, probabilistic history-based methods directly predict the distribution over future trajectories, mitigating the sampling overhead. However, this approach make the learning problem hard as the model needs to learn the future distribution for multiple time steps ahead. To account for this, complex models (e.g. large neural networks) are often necessary. This can prevent their usage in embedded systems with limited memory capacity.

In this work, we present a novel approach for modeling stochastic dynamical systems with interacting agents that is able to generate expressive multi-modal predictive distributions over future trajectories in a runtime and memory efficient manner. We model the unknown stochastic dynamical system as a *Deep State-Space Model* (DSSM) in which the shared dynamics of all agents are modeled in a joint latent space using GNNs. Our model belongs to the family of recurrent neural networks and we replace the expensive MC operations during training and testing by introducing a novel deterministic moment matching scheme. Prior work (Look et al., 2022) on moment matching for dynamical systems does not consider interacting systems and is restricted to unimodal processes. We overcome the first limitation by applying GNNs in the transition model and the second limitation by placing a *Gaussian Mixture Model* (GMM) over the initial latent state.

For each mixture component, we independently apply our moment matching rules in order to arrive at multimodal predictive distributions over future trajectories. In autonomous driving, the initial latent state is often estimated from historical information and can provide information about the drivers' intentions (Tang & Salakhutdinov, 2019). Conditioned on the initial latent state, the predictive distribution can often be accurately modeled with an unimodal distribution (Cui et al., 2019; Chai et al., 2019). Finally, as there exists a wide variety of dense traffic scenarios, the high number of agents can result in prohibitively large GMM covariance matrices as their size grows quadratically with the number of traffic participants. We address this problem by proposing structured covariance approximations.

We summarize our contribution as follows:

- We derive output moments for GNN layers, which makes GNNs applicable to moment matching algorithms. This leads to the first deterministic inference scheme for deep state-space models for interacting systems.

- We introduce a GMM distribution over the initial latent states that results in multimodal predictive distributions over future trajectories.

- We propose structured approximations to the GMM covariance matrices that can reduce the computational complexity of our approach from $O(M^3)$ to $O(M^2)$, where $M$ is the number of agents.

In our experiments, we benchmark our proposed model on two challenging autonomous driving datasets. Our results demonstrate that our deterministic model has strong empirical performance compared to state-of-the-art alternatives. We visualize the predictive output distribution for a real-world traffic scenario on a roundabout with multiple agents in Fig. 1. The future distribution is highly multi-modal, as traffic participants can leave the roundabout from several exits. Our model is capable of predicting multiple modes, which we efficiently approximate as a GMM, and takes interactions into account using GNNs.

To gain further insights into our model and inference scheme, we carefully examine the impact of the individual contributions of our work in an ablation study. Next, we provide an empirical runtime study of our covariance approximations. Our findings indicate that sparse covariance approximations reduce the computational complexity by a factor of up to 100, which makes them favourable for applications with limited computational resources. We conclude our experiments by studying the generalization capabilities of our model on out-of-distribution data, e.g. traffic environments that have not been observed during training.

## 2 Background

In this chapter, we first provide background on (deep) state-space models for single-agent systems. We proceed by giving a small recap on moment matching rules that allow us to propagate the latent dynamics forward in time in a deterministic manner. Next, we review graph neural networks for interaction modeling. State-space models and graph neural networks form the basis for our new model for stochastic dynamical interacting systems, which we introduce in Sec. 3. The deterministic moment matching rules build the foundation of our training and testing routines that we describe in Sec. 4.

### 2.1 Deep State-Space Models

*State-Space Models* (SSM) are a model class for dynamical systems (e.g. Schön et al. (2011), Särkkä (2013)) that assume that each $D_y$-dimensional observed variable $y_t \in \mathbb{R}^{D_y}$ is emitted by a latent $D_x$-dimensional latent variable $x_t \in \mathbb{R}^{D_x}$. The latents are coupled via first-order Markovian dynamics, e.g. the state at time point $x_t$ only depends on the state of the previous time point $x_{t-1}$. Typically the observed state $y_{t-1}$ does not contain all necessary information in order to reliably predict the next observed state $y_t$. Consider the case of traffic forecasting in which the observed state $y_t$ contains the position of a vehicle but not its velocity or acceleration data. We can then supply the latent state $x_t$ with the missing information in order to allow for accurate forecasts about the next time point. Consequently, SSMs are a flexible model class that allows us to make reliable forecasts about complex systems.

A *Deep State-Space Model* (DSSM) is a non-linear SSM in which the transition model, that maps the latent state from the current to the next time point, and the emission model, that maps the latent state to the outputs, are realized by neural networks. They come in handy for applications in which the true underlying dynamics have an unknown non-linear functional form and must be estimated from data. Assuming additive Gaussian noise (e.g. Krishnan et al. (2017)), their generative model can be written down as follows

$$x_0 \sim p(x_0|\mathcal{I}), \tag{1}$$
$$x_t \sim \mathcal{N}\left(x_t|x_{t-1} + f(x_{t-1}, \mathcal{I}), \text{diag}\left(L(x_{t-1}, \mathcal{I})\right)\right), \qquad t = 1, \ldots, T \tag{2}$$
$$y_t \sim \mathcal{N}\left(y_t|g(x_t), \text{diag}\left(\Gamma(x_t)\right)\right), \qquad t = 1, \ldots, T \tag{3}$$

where $\mathcal{I} \in \mathbb{R}^{D_\mathcal{I}}$ is the context variable that encodes auxiliary information, such as historical or relational information. The mean update $f(x_t, \mathcal{I}) : \mathbb{R}^{D_x} \times \mathbb{R}^{D_\mathcal{I}} \to \mathbb{R}^{D_x}$, governing the deterministic component of the transition model, is parameterized by a neural net with an arbitrary architecture. Similarly, the variance update $L(x_t, \mathcal{I}) : \mathbb{R}^{D_x} \times \mathbb{R}^{D_\mathcal{I}} \to \mathbb{R}^{D_x}_+$ is parameterized by another neural net, which models the stochasticity of the system. Both $f$ and $L$ are neural networks, which are parameterized by $\theta_f$ and $\theta_L$. The emission model follows a Gaussian distribution with mean $g(x_t) : \mathbb{R}^{D_x} \to \mathbb{R}^{D_y}$ and variance $\Gamma(x_t) : \mathbb{R}^{D_x} \to \mathbb{R}^{D_y}_+$, where $g$ and $\Gamma$ are both neural networks with arbitrary architecture and parameters $\theta_g$ and $\theta_\Gamma$.

Assuming additive Gaussian noise allows us to interpret the transition model [Eq. (2)] as a discretized neural stochastic differential equation (Tzen & Raginsky, 2019; Look et al., 2022). While we do not pursue this line of work any further, we note that this connection allows for straight-forward extensions to irregularly sampled time series. Finally, there exists work that couples state-space models with recurrent neural networks (Chung et al., 2015; Fraccaro et al., 2016) whose gating mechanism can help in learning long-term effects.

## 2.2 Transition Kernel

In this section, we provide background on how to compute the $t$-step transition kernel, $p(x_t|x_0, \mathcal{I})$ with $t > 1$, which allows us to propagate the latent state forward in time for general state-space models. It is defined by the following recurrence

$$p(x_t|x_0, \mathcal{I}) = \int p(x_t|x_{t-1}, \mathcal{I})p(x_{t-1}|x_0, \mathcal{I})dx_{t-1}, \tag{4}$$

where $p(x_0|\mathcal{I})$ follows Eq. (1). It is worth noting that Eq. (4) cannot be computed in closed-form since the distribution $p(x_{t-1}|x_0, \mathcal{I})$ has to be propagated through the non-linear transition model $p(x_t|x_{t-1}, \mathcal{I})$ [Eq. 2].

### 2.2.1 Overview

Various approximations to the transition kernel [Eq. (4)] have been proposed that can be roughly split into two groups: (a) MC sampling based approaches (Brandt & Santa-Clara, 2002; Pedersen, 1995; Elerian et al., 2001) and (b) deterministic approximations based on assumed densities (Särkkä et al., 2015). While MC based approaches can, in the limit of infinitely many samples, approximate arbitrarily complex distributions, they are often slow in practice and their convergence is difficult to assess. In contrast, deterministic approaches often build on the assumption that the $t$-step transition kernel can be approximated by a Gaussian distribution. This assumption can be justified if the transition model can be locally linearly approximated and the observations are sufficiently densely sampled. The transition kernel is then computed in an iterative manner by applying a Gaussian approximation at each time step and propagating the moments along the time direction using a numerical integration scheme (Särkkä & Sarmavuori, 2013; Särkkä et al., 2015).

Prior work in the context of neural SDEs (Look et al., 2022) proposes to first discretize the differential equations and afterwards apply moment matching through the neural network layers as numerical integration scheme. Since the resulting algorithm propagates moments through time and neural network layers, it is termed *Bidimensional Moment Matching* (BMM). This approach was shown to be superior over standard numerical integration schemes in terms of compute and accuracy. Since the transition model in SSMs can be interpreted as discretized SDEs (Särkkä et al., 2015), we build our work on their approach. Moment propagation through neural network layers has also been applied previously in the context of expectation

propagation (Hernandez-Lobato & Adams, 2015; Ghosh et al., 2016), deterministic variational inference (Wu et al., 2019), and evidential deep learning (Haussmann et al., 2020).

### 2.2.2 Bidimensional Moment Matching

In this section, we recapitulate the original *Bidimensional Moment Matching* (BMM) algorithm of Look et al. (2022) and its use for propagating the latent state forward in time [Eq. (2)]. BMM approximates the transition kernel $p(x_t|x_0, \mathcal{I})$ by combining horizontal moment matching along the time axis with vertical moment matching across the neural network layers.

**Horizontal Moment Matching**  In order to facilitate the computation of the transition kernel, we replace $p(x_t|x_0, \mathcal{I})$ for all time steps $t = 1, \ldots, T$ with a Gaussian distribution

$$p(x_t|x_0, \mathcal{I}) = \int p(x_t|x_{t-1}, \mathcal{I}) p(x_{t-1}|x_0, \mathcal{I}) dx_{t-1} \tag{5}$$
$$\approx \mathcal{N}(x_t|\mu_t(\mathcal{I}), \Sigma_t(\mathcal{I})),$$

with mean $\mu_t(\mathcal{I})$ and covariance $\Sigma_t(\mathcal{I})$. This approximation simplifies the problem to calculating the first two moments of the transition kernel and is assumed to work well if the dynamics can be locally approximated by a linear model, which is the case for many applications. Assuming that the one-step transition kernel $p(x_t|x_{t-1}, \mathcal{I})$ follows Eq. (2), the mean $\mu_t(\mathcal{I})$ and covariance $\Sigma_t(\mathcal{I})$ can be computed as a function of prior moments $\mu_{t-1}(\mathcal{I})$ and $\Sigma_{t-1}(\mathcal{I})$ (Look et al., 2022)

$$\mu_t(\mathcal{I}) = \mu_{t-1}(\mathcal{I}) + \mathbb{E}[f(x_{t-1}, \mathcal{I})], \tag{6}$$
$$\Sigma_t(\mathcal{I}) = \Sigma_{t-1}(\mathcal{I}) + \mathrm{Cov}[f(x_{t-1}, \mathcal{I})] + \mathrm{Cov}[x_{t-1}, f(x_{t-1}, \mathcal{I}),] + \mathrm{Cov}[x_{t-1}, f(x_{t-1}, \mathcal{I})]^T + \mathrm{diag}\left(\mathbb{E}[L(x_{t-1}, \mathcal{I})]\right),$$

where $\mathrm{Cov}[x_{t-1}, f(x_{t-1}, \mathcal{I})]$ denotes the cross-covariance between the random vectors in the arguments.

**Vertical Moment Matching**  The mean $\mathbb{E}[f(x_{t-1}, \mathcal{I})]$ and covariance $\mathrm{Cov}[f_\theta(x_{t-1}, \mathcal{I})]$ of the transition function, as well as the expected variance update $\mathbb{E}[L(x_{t-1}, \mathcal{I})]$, can be computed as a result of moment propagation through neural network layers. For many common layers, including affine transformations and ReLU activation functions, the corresponding output moments can be either computed in closed-form or good approximations are available in the literature (Wu et al., 2019). In contrast, approximating the cross-covariance $\mathrm{Cov}[x_t, f_\theta(x_t, \mathcal{I})]$ cannot be achieved using moment matching rules since we cannot decompose the cross-covariance term into layerwise operations. Instead, we resort to Stein's Lemma using

$$\mathrm{Cov}[x_t, f(x_t, \mathcal{I})] = \mathrm{Cov}[x_t]\mathbb{E}[\nabla_{x_t} f(x_t, \mathcal{I})], \tag{7}$$

where the expected Jacobian can be approximated as (Look et al., 2022)

$$\mathbb{E}[\nabla_{x_t} f(x_t, \mathcal{I})] \approx \prod_{l=1}^{L} \mathbb{E}[J_t^l]. \tag{8}$$

Above, $J_t^l$ denotes the Jacobian of layer $l$ at time step $t$. The expectation of the Jacobian is analytically available or can be closely approximated for common layer types.

### 2.3 Graph Neural Networks

*Graph neural networks* (GNNs) have emerged as a powerful method for interaction modeling (Battaglia et al., 2018; Hamilton et al., 2017; Gilmer et al., 2017). Given a set of agents and relational information in form of a graph, each agent corresponds to one node in the graph that is equipped with a set of features. The relation between the agents is encoded via the edges and information exchange between the agents takes place by sending messages along the edges. By performing multiple rounds of message-passing, information can flow along the graph. This allows for interactions between non-adjacent agents, provided that a path between the agents exists.

More formally, we define the structure of our GNN as follows. For $M$ agents, a GNN receives as inputs a set of node features $x = \{x^m\}_{m=1}^M$, where $x \in \mathbb{R}^{MD_x}$ and $x^m \in \mathbb{R}^{D_x}$, and a set of edges $\mathcal{E} = \{e^{m,m'}\}_{m,m'=1}^M$ which is part of the context variable $\mathcal{I} \in \mathbb{R}^{D_\mathcal{I}}$. The edge attribute $e^{m,m'}$ has a binary encoding, where $e^{m,m'} = 1$ if agent $m$ and agent $m'$ are related. The GNN output is an update of the node features, i.e. $z = \text{GNN}(x, \mathcal{I})$ with $z \in \mathbb{R}^{MD_z}$, and consists of the following two steps that may be repeated multiple times:

1. For each agent $m$, receive message $x^{\mathcal{N}_m} \in \mathbb{R}^{D_x}$ by aggregrating information from neighboring agents:

$$x^{\mathcal{N}_m} = \text{AGG}\left(\{x^{m'}|e^{m,m'} = 1\}\right) \in \mathbb{R}^{D_x}, \tag{9}$$

where $\{x^{m'}|e^{m,m'} = 1\}$ denotes the set of all neighbours of node $m$. The aggregation operation AGG is permutation invariant, i.e. it does not change when the ordering of the inputs is swapped and generalizes to a varying number of inputs. A commonly used aggregation operation that we also apply in our work is the mean function.

2. For each agent $m$, update the node information:

$$z^m = \text{UPDATE}(x^m, x^{\mathcal{N}_m}, \mathcal{I}), \tag{10}$$

where $\text{UPDATE}(x^m, x^{\mathcal{N}_m}, \mathcal{I}) : \mathbb{R}^{D_x} \times \mathbb{R}^{D_x} \times \mathbb{R}^{D_\mathcal{I}} \to \mathbb{R}^{D_z}$ is typically implemented by a neural network.

A simple form of an interacting dynamical system takes the features of each agent at its current position and connects agents that are within a pre-defined radius with edges. The GNN operation updates the position and velocity information of each agent by taking information of the adjacent traffic participants into account.

## 3 Graph Deep State-Space Models

We aim to model stochastic dynamical interactions between agents following complex behavioural patterns, such as road traffic interactions. We extend deep state-space models to interacting systems by proposing *Graph Deep State-Space Models* (GDSSM), which use graph neural networks in the transition model to capture interactions between agents. We define our probabilistic model in this section and then introduce a novel scheme for efficient and deterministic training and predictions in the subsequent section.

We are interested in modeling the dynamics of $M$ interacting agents with deep state-space models by using a coupled latent space. In other words, instead of using a $D_x$-dimensional latent space for each agent, we assume that the agents share a latent space of size $MD_x$. Since (i) the number of agents can vary between scenes, (ii) the transition model should be agnostic to the order of the agents and (iii) it is challenging to parameterize high-dimensional latent spaces, we use GNNs in the transition model.

More formally, we denote the state of agent $m$ at time step $t$ as $x_t^m \in \mathbb{R}^{D_x}$ and the set of all state variables as $x_t = \{x_t^m\}_{m=1}^M$. The dynamics of $x_t$ follow Eq. (2), where the mean update $f(x_t, \mathcal{I}) : \mathbb{R}^{MD_x} \times \mathbb{R}^{D_\mathcal{I}} \to \mathbb{R}^{MD_x}$ and the variance update $L(x_t, \mathcal{I}) : \mathbb{R}^{MD_x} \times \mathbb{R}^{D_\mathcal{I}} \to \mathbb{R}^{MD_x}$ are implemented with the help of graph neural networks

$$f(x_t, \mathcal{I}) = \begin{bmatrix} \tilde{f}(x_t^1, x_t^{\mathcal{N}_1}, \mathcal{I}) \\ \vdots \\ \tilde{f}(x_t^M, x_t^{\mathcal{N}_M}, \mathcal{I}) \end{bmatrix}, \qquad L(x_t, \mathcal{I}) = \begin{bmatrix} \tilde{L}(x_t^1, x_t^{\mathcal{N}_1}, \mathcal{I}) \\ \vdots \\ \tilde{L}(x_t^M, x_t^{\mathcal{N}_M}, \mathcal{I}) \end{bmatrix}. \tag{11}$$

The agent-specific mean update is denoted by $\tilde{f}(x_t^m, x_t^{\mathcal{N}_m}, \mathcal{I}) : \mathbb{R}^{D_x} \times \mathbb{R}^{D_x} \times \mathbb{R}^{D_\mathcal{I}} \to \mathbb{R}^{D_x}$ and the variance by $\tilde{L}(x_t^m, x_t^{\mathcal{N}_m}, \mathcal{I}) : \mathbb{R}^{D_x} \times \mathbb{R}^{D_x} \times \mathbb{R}^{D_\mathcal{I}} \to \mathbb{R}_+^{D_x}$. Both implement the update function in general graph neural networks [Eq. (10)], whereas , $x_t^{\mathcal{N}_m}$ contains aggregated information of the states from all neighboring agents [Eq.(9)]. A deterministic variant of our model, e.g. setting $L(x_t, \mathcal{I}) = 0$, has been successfully used for

learning surrogate models for complex physical systems (Sanchez-Gonzalez et al., 2020). We further assume that it is sufficient to couple the latent dynamics across the agents and keep the emission model [Eq. (3)] independent across agents. Note that our transition model consists of a single aggregation and update step which is sufficient if the data is densely sampled such that the information flow between agents is fast compared to the evolution of the state dynamics. However, it is also straight-forward to extend the model to multiple message-passing steps per time point by stacking multiple GNN layers in the mean and variance update function.

Furthermore, we note that although the transition noise factorizes across agents, correlations between agents emerge since the mean and the variance depend not only on the state of the $m$-th agent, but also on the states of all neighboring agents. After $a$ aggregation steps, our GNN model accounts for correlations between agent $m$ and agent $m'$ provided that they are connected by a path that is at most $a$ steps long. In contrast, methods, that only take the state of the $m$-th agent into account, do not lead to any correlations, while methods, that take the state of all other agents into account, lead to a fully correlated covariance matrix after one time step.

In order to complete the probabilistic description of our model, we further specify the distribution of the initial latent state $x_0 \in \mathbb{R}^{MD_x}$ with a *Gaussian Mixture Model* (GMM)

$$
\begin{aligned}
v &\sim \mathrm{Cat}([\pi_1(\mathcal{I}), \ldots, \pi_V(\mathcal{I})]), \\
x_0 &\sim \mathcal{N}(x_0 | \mu_{0,v}(\mathcal{I}), \mathrm{diag}(\Sigma_{0,v}(\mathcal{I}))),
\end{aligned}
\tag{12}
$$

where $\mathrm{Cat}([\pi_1(\mathcal{I}), \ldots, \pi_V(\mathcal{I})])$ is a categorical distribution with $V$ mixture components. Each component is specified by its weight $\pi_v(\mathcal{I}) : \mathbb{R}^{D_\mathcal{I}} \to \mathbb{R}_+$, mean $\mu_{0,v}(\mathcal{I}) : \mathbb{R}^{D_\mathcal{I}} \to \mathbb{R}^{MD_x}$, and diagonal covariance $\Sigma_{0,v}(\mathcal{I}) : \mathbb{R}^{D_\mathcal{I}} \to \mathbb{R}_+^{MD_x}$. The weights $\pi_{1:V}$ form a standard V-simplex. We use a GNN, which we refer to as the embedding function $\mathrm{h}(\mathcal{I}) : \mathbb{R}^{D_\mathcal{I}} \to \mathbb{R}^{V+2VMD_x}$, in order to model the initial state distribution

$$
\mathrm{h}(\mathcal{I}) = \begin{bmatrix} \pi_{1:V}(\mathcal{I}) \\ \mu_{0,1:V}(\mathcal{I}) \\ \Sigma_{0,1:V}(\mathcal{I}) \end{bmatrix}.
\tag{13}
$$

We assume that the context variable $\mathcal{I}$ contains relational information as a set of edges as well as historical information for each agent in the form of an observed trajectory. In a sense, the embedding function acts hereby as a filter, which learns a distribution over the initial latent state from past observations.

In autonomous driving, the initial latent state can be connected to the drivers' intention (Tang & Salakhutdinov, 2019). The context information $\mathcal{I}$ is often not sufficient to rule out different hypotheses about the future, e.g. does the car behind us want to overtake in the next five seconds or not. Using a mixture model allows us to incorporate different hypotheses into the model in a principled manner, which will ultimately lead to highly multimodal predictive distributions.

State-space models and graph neural networks have been previously combined for multi-agent trajectory forecasting by Yang et al. (2020). In contrast to our work, the authors (i) use a different model definition by applying recurrent neural networks and a non-Gaussian density in the transition model and (ii) perform Monte Carlo sampling during inference which can lead to slow convergence. In the next chapter, we show that our model definition allows for more efficient training by performing deterministic moment matching rules. We compare against Monte Carlo alternatives in our experiments.

## 4 Deterministic Approximations for GDSSMs

In this chapter, we present our novel inference scheme for GDSSMs that enables efficient and deterministic training and predictions. In Sec. 4.1, we first give a short overview over existing inference techniques and compare it to our scheme which aims at directly maximizing the predictive log-likelihood of future trajectories. The predictive log-likelihood cannot be computed in closed-form. We propose an efficient and deterministic approximation to it in Sec. 4.2 that relies on bidimensional moment matching. Our moment matching rules necessitate the computation of output moments and expected Jacobians of graph neural

network layers. We present ouput moments for commonly used layers in Sec. 4.3. As our algorithm approximates the output distribution at each time step with a Gaussian mixture distribution over all agents, the resulting covariance matrix for each mixture component can become computationally intractable for a large number of traffic participants. We address this pain point in Sec. 4.4 by proposing sparse approximations to the covariance.

## 4.1 Parameter Inference

Classical inference methods for state-space models aim at directly maximizing the log-likelihood of the data

$$\log p(y_1, \ldots, y_T | \mathcal{I}) = \log \int p(x_0|\mathcal{I}) \prod_{t=1}^{T} p(x_t|x_{t-1}, \mathcal{I}) p(y_t|x_t) dx_0 \ldots dx_T, \tag{14}$$

where $p(x_t|x_{t-1}, \mathcal{I})$ is defined in Eq. (2) and $p(y_t|x_t)$ in Eq. (3). This quantity can only be computed in closed-form if the emission and transition model are linear Gaussians. In our case, the transition model is parameterized by a graph neural network and the emission model by a standard neural network. Both functions are highly non-linear and render an analytical solution to Eq. (14) infeasible.

Therefore, most existing methods apply either a particle filter (Schön et al., 2015), variational inference (Krishnan et al., 2017; Bayer et al., 2021) or a combination of both (Naesseth et al., 2018; Maddison et al., 2017; Le et al., 2018) in order to approximate the log-likelihood. All of these approaches have in common that they require learning a proposal distribution $q(x_0, \ldots, x_T | y_1, \ldots, y_T)$: In particle filtering, the proposal distribution is recursively defined by the importance function $q(x_t|x_0, \ldots, x_{t-1}, y_1, \ldots, y_t)$ and is optimal in terms of variance by setting $q(x_t|x_0, \ldots, x_{t-1}, y_1, \ldots, y_t) = p(x_t|x_{t-1}, y_t)$ (e.g. Doucet et al. (2000). Variational inference aims to find the best approximation to the true posterior within the chosen variational family by minimizing the *Kullback-Leibler* (KL) divergence between the true and approximate posterior (Blei et al., 2017), while variational sequential Monte Carlo seeks to minimize the KL divergence on an extended sampling space (Le et al., 2018).

However, the proposal distribution is only used as an auxiliary tool during inference. For many prediction tasks (Djuric et al., 2020; Jain et al., 2019; Chai et al., 2019), the quantity of interest is the *predictive log-likelihood* (PLL)

$$\begin{aligned} \text{PLL}(y_1, \ldots, y_T | \mathcal{I}) &= \sum_{t=1}^{T} \log p(y_t | \mathcal{I}) \\ &= \sum_{t=1}^{T} \log \int p(x_0|\mathcal{I}) p(x_t|x_0, \mathcal{I}) p(y_t|x_t) dx_0 dx_t, \end{aligned} \tag{15}$$

where the transition kernel $p(x_t|x_0, \mathcal{I})$ is defined in Eq. (4).

In contrast to the standard training objective, the predictive log-likelihood propagates the latent state forward in time without receiving any feedback from the observations; mimicking the behavior during test time. Since we want to use the same objective during training and test time, we opt for directly maximizing the predictive log-likelihood during training. The observation that using one-step ahead predictions in training is not sufficient in order to obtain reliable multi-step ahead predictions during testing has also been made in Bengio et al. (2015) where the authors propose a scheduled sampling strategy in order to gradually switch from single to multi-step ahead predictions during training. Multi-step ahead training has also been successfully applied for spatio-temporal forecasting(Pal et al., 2021) and model-based planning (Hafner et al., 2019).

The PLL aims at optimizing the average marginal log-likelihood $\log p(y_t|\mathcal{I})$, while the standard objective aims at optimizing the joint likelihood $\log p(y_1, \ldots, y_T|\mathcal{I})$. We deem the choice between the two objectives application dependent: The PLL is most useful for tasks that can be solved by assessing the marginal distributions only. An important and large application class that falls into this category is in the context of autonomous driving in which the marginal distributions of neighboring traffic participants at a given time

horizon are often sufficient in order to control the car (e.g. Herman et al. (2022)). As a consequence, many papers in the autonomous driving literature report as evaluation metrics the performance of their method at fixed time intervals which matches our PLL objective (e.g. Djuric et al. (2020); Jain et al. (2019); Chai et al. (2019)). In contrast, optimizing the joint log-likelihood is preferred for tasks that require sampling realistic looking trajectories, as it is done for instance in sentence generation (Vaswani et al., 2017).

### 4.2 Approximating the Predictive Log-Likelihood using Bidimensional Moment Matching

We are interested in the predictive log-likelihood $\text{PLL}(y_1, \ldots, y_T | \mathcal{I})$, which describes the predictive log-likelihood of all traffic participants up to time step $T$. In order to calculate it, we need to solve the nested set of integrals given in Eq. (15). The BMM algorithm allows us to approximate the transition kernel $p(x_t | \mathcal{I}) = \int p(x_t | x_0, \mathcal{I}) p(x_0 | \mathcal{I}) dx_0$ in case that the initial state $x_0$ has a Gaussian distribution. However, in our model formulation the initial latent state $x_0$ follows a GMM to allow for multimodality. In order to account for that, we approximate the marginal latent distribution $p(x_t | \mathcal{I})$ as

$$p(x_t | \mathcal{I}) \approx \sum_{v=1}^{V} \pi_v(\mathcal{I}) p(x_{t,v} | \mathcal{I}), \tag{16}$$

where each mixture component $p(x_{t,v} | \mathcal{I})$ is approximated with the BMM algorithm as $p(x_{t,v} | \mathcal{I}) \approx \mathcal{N}(\mu_{t,v}(\mathcal{I}), \Sigma_{t,v}(\mathcal{I}))$. Assuming a GMM at the initial state allows us to efficiently obtain multimodal predictions while being computationally efficient: We obtain multimodal distributions by explicitly marginalizing over the mixture components and we compute each component efficiently by applying the BMM algorithm. Finally, we approximate the marginal distribution $p(y_t | \mathcal{I})$ as a GMM by another round of moment matching

$$p(y_t | \mathcal{I}) = \int p(y_t | g(x_t), \text{diag}(\Gamma(x_t))) p(x_t | \mathcal{I}) dx_t \tag{17}$$

$$\approx \sum_{v=1}^{V} \pi_v(\mathcal{I}) \mathcal{N}(a_{t,v}(\mathcal{I}), B_{t,v}(\mathcal{I})),$$

where $a_{t,v}(\mathcal{I})$ and $B_{t,v}(\mathcal{I})$ are the mean and covariance of the $v$-th mixture component at the $t$-th time step. These two moments are available as

$$a_{t,v}(\mathcal{I}) = \mathbb{E}[g(x_{t,v})], \qquad\qquad B_{t,v}(\mathcal{I}) = \text{Cov}[g(x_{t,v})] + \text{diag}\left(\mathbb{E}[\Gamma(x_t)]\right), \tag{18}$$

which is a direct outcome of the law of the unconscious statistician. We present the pseudocode for computing $p(y_t | \mathcal{I})$ using our method in Algorithm 1. Algorithm 2 further shows how we can optimize our model parameters employing the PLL [Eq. (15)] as a training objective.

In our method, we approximate $p(x_{t,v} | \mathcal{I})$ and $p(y_t | \mathcal{I})$ by applying moment matching to each Gaussian mixture component independently. We note that this approximation becomes exact for locally linear transition and emission functions as stated in Thm. 1.

**Theorem 1.** *The marginal distribution $p(y_t | \mathcal{I})$ is analytically computed as*

$$p(y_t | \mathcal{I}) = \sum_{v=1}^{V} \pi_v(\mathcal{I}) \mathcal{N}(a_{t,v}(\mathcal{I}), B_{t,v}(\mathcal{I})),$$

*for a GDSSM with the below generative model*

$$
\begin{aligned}
v &\sim Cat([\pi_1(\mathcal{I}), \ldots, \pi_V(\mathcal{I})]), \\
x_0 &\sim \mathcal{N}(\mu_{0,v}(\mathcal{I}), diag(\Sigma_{0,v}(\mathcal{I}))), \\
x_t &\sim \mathcal{N}\left(x_t | x_{t-1} + f(t, v, \mathcal{I}) x_{t-1}, diag\left(L(t, v, \mathcal{I})\right)\right), && t = 1, \ldots, T \\
y_t &\sim \mathcal{N}\left(y_t | g(t, v, \mathcal{I}) x_t, diag\left(\Gamma(t, v, \mathcal{I})\right)\right), && t = 1, \ldots, T
\end{aligned}
$$

*where $f(t, v, \mathcal{I}), L(t, v, \mathcal{I}), g(t, v, \mathcal{I}), \Gamma(t, v, \mathcal{I})$ are time $t$, component $v$, and context $\mathcal{I}$ depending matrices with appropriate dimensionality.*

*Proof.* The proof is straightforward as the output moments of the transition and emission function are analytically tractable. We provide details in App. A. $\qquad\square$

---

**Algorithm 1** Bidimensional Moment Matching in Latent Space (BMMLS)

---

**Inputs:** $T$ ▷ Prediction horizon
$\qquad\quad\mathcal{I}$ ▷ Context variable
$\qquad\quad f(x_t, \mathcal{I}; \theta_f)$ ▷ Mean update
$\qquad\quad L(x_t, \mathcal{I}; \theta_L)$ ▷ Covariance update
$\qquad\quad g(x_t; \theta_g)$ ▷ Mean emission
$\qquad\quad \Gamma(x_t; \theta_\Gamma)$ ▷ Covariance emission
$\qquad\quad h(\mathcal{I}; \theta_h)$ ▷ Embedding function
**Outputs:** Approximate marginal distribution $p(y_T|\mathcal{I})$

$\mu_{0,1:V}(\mathcal{I}), \Sigma_{0,1:V}(\mathcal{I}), \pi_v(\mathcal{I}) = h(\mathcal{I}; \theta_h)$ ▷ GMM at initial step, Eq. 13
**for** mixture component $v \in \{1, \cdots, V\}$ **do**
$\quad$ **for** time step $t \in \{0, \cdots, T-1\}$ **do** ▷ Horizontal Moment Matching
$\qquad \mu_{t+1,v} \leftarrow \mu_t(\mathcal{I}) + \mathbb{E}[f(x_t, \mathcal{I}; \theta_f)]$ ▷ Eq. 6
$\qquad \Sigma_{t+1,v} \leftarrow \Sigma_t(\mathcal{I}) + \text{Cov}[f(x_t, \mathcal{I}; \theta_f)] + \text{Cov}[x_t, f(x_t, \mathcal{I}; \theta_f),] + \text{Cov}[x_t, f(x_t, \mathcal{I}; \theta_f)]^T + \text{diag}\left(\mathbb{E}[L(x_t, \mathcal{I}; \theta_L)]\right)$ ▷ Eq. 6
$\quad$ **end for**
$\quad a_{T,v}(\mathcal{I}) \leftarrow \mathbb{E}[g(x_{T,v}; \theta_g)]$ ▷ Eq. 18
$\quad B_{T,v}(\mathcal{I}) \leftarrow \text{Cov}[g(x_{T,v}; \theta_g)] + \text{diag}\left(\mathbb{E}[\Gamma(x_T; \theta_\Gamma)]\right)$ ▷ Eq. 18
**end for**
**return** $\sum_{v=1}^{V} \pi_v(\mathcal{I})\mathcal{N}(a_{T,v}(\mathcal{I}), B_{T,v}(\mathcal{I}))$

---

**Algorithm 2** Deterministic Training of a GDSSM

---

**Inputs:** $\mathcal{D}$ ▷ Dataset
$\qquad\quad f(x_t, \mathcal{I}; \theta_f)$ ▷ Mean update
$\qquad\quad L(x_t, \mathcal{I}; \theta_L)$ ▷ Covariance update
$\qquad\quad g(x_t; \theta_g)$ ▷ Mean emission
$\qquad\quad \Gamma(x_t; \theta_\Gamma)$ ▷ Covariance emission
$\qquad\quad h(\mathcal{I}; \theta_h)$ ▷ Embedding function
**Outputs:** Optimized weights $\theta = \{\theta_f, \theta_L, \theta_g, \theta_\Gamma, \theta_h\}$

**while** not converged **do**
$\quad \mathcal{I}, y_{1:T} \sim D$ ▷ Sample $\mathcal{I}$ and $y_{1:T}$ from dataset
$\quad$ **for** $y_t \in y_{1:T}$ **do**
$\qquad p(y_t|\mathcal{I}) = \text{BMMLS}(t, \mathcal{I}, f, L, g, \Gamma, h)$ ▷ Approximate marginal distribution $p(y_t|\mathcal{I})$ via algorithm 1
$\quad$ **end for**
$\quad \text{PLL}(y_1, \ldots, y_T|\mathcal{I}) = \sum_{t=1}^{T} \log p(y_t|\mathcal{I})$ ▷ Calculate predictive log-likelihood 15
$\quad \theta \leftarrow \theta + \frac{\partial PLL(y_{1:T}|\mathcal{I})}{\partial \theta}$ ▷ Update weights $\theta$
**end while**
**return** $\theta$

---

For general non-linear transition and emission functions, the quality of this scheme depends on two factors: (a) how well the transition and emission function can be approximated in a locally linear fashion and (b) if the covariance matrix $\Sigma_{t,v}$ is small enough over the time horizon $t$. We observe in our experiments that the approximation works well and further illustrate its behavior on a small toy dataset in Fig. 2. Finally, we note that Gaussian mixture models have also been successfully used for filtering problems (Alspach & Sorenson, 1972) where, similar as in our work, moment matching is performed for each component individually in the predict step.

## 4.3 Output Moments of Graph Neural Network Layers

In order to use the the BMM framework, we need to be able to calculate the first two output moments as well as the expected Jacobian of graph neural network layers. In the following, we derive the analytic expression of the output moments and expected Jacobian for the common graph neural net layers: (i) node-wise affine transformation, and (ii) mean aggregation. Output moments for the ReLU activation are provided in Wu et al. (2019). For completeness, we also provide the output moments for the ReLU activation in App. B.

Let $x_t^{l,m} \in \mathbb{R}^{D_{x,l}}$ be the node features at layer $l$ of node $m$ at time step $t$ and $x_t^l = \{x_t^{l,m}\}_{m=1}^{M}$ the set of all node features with $x_t^l \in \mathbb{R}^{MD_{x,l}}$. For the sake of brevity, we have omitted here the index of the mixture

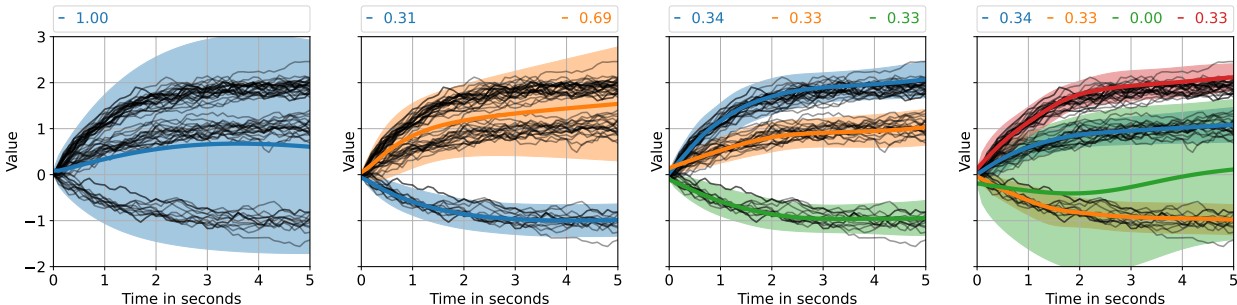

Figure 2: Predictions of our model after training on an one-dimensional, multi-modal toy problem. Solid black lines represent the ground truth dynamics with three different modes of equal probability. For our model, we report for each predicted mode the mean, 95% confidence interval and mixture weight. We set the dimension of the latent space to $D_x = 3$ and vary (from left to right) the number of modes $V$ in the model from 1 to 4. For $V < 3$, the number of modes is set too small, and we observe mode covering behavior. For $V \geq 3$, our model can recover the ground truth dynamics. If the number of components is too high (V=4), the mixture weights of redundant components are set to zero.

component $v$. We denote mean and covariance of a graph with $M$ nodes at layer $l$ and time step $t$ as

$$
\mathbb{E}[x_t^l] = \begin{bmatrix} \mathbb{E}[x_t^{l,1}] \\ \vdots \\ \mathbb{E}[x_t^{l,M}] \end{bmatrix}, \qquad \mathrm{Cov}[x_t^l] = \begin{bmatrix} \mathrm{Cov}[x_t^{l,1}, x_t^{l,1}] & \dots & \mathrm{Cov}[x_t^{l,1}, x_t^{l,M}] \\ \vdots & \ddots & \vdots \\ \mathrm{Cov}[x_t^{l,M}, x_t^{l,1}] & \dots & \mathrm{Cov}[x_t^{l,M}, x_t^{l,M}]. \end{bmatrix}, \tag{19}
$$

where $\mathbb{E}[x_t^{l,m}] \in \mathbb{R}^{D_{x,l}}$ and $\mathrm{Cov}[x_t^{l,m}, x_t^{l,m'}] \in \mathbb{R}^{D_{x,l} \times D_{x,l}}$. A typical GNN architecture (see Sec. 2.3) consists of alternating aggregation steps, in which information of all neighbors is collected, and update steps, in which the features of the node are updated. For the aggregation step, we derive the output moments for the commonly used mean aggregation operation in Sec. 4.3.3. For the update step, we assume that the neural network is built as a sequence of affine transformations and nonlinearities. The output moments of nonlinear activations are applied independently across agents. As a consequence, their rules do not change when used in the GNN setting and we can use the derivations from Wu et al. (2019). Hence it remains open to derive the output moments for affine transformations, as they are used in GNN context, which we tackle in Sec. 4.3.2. The mean aggregation operation and the node-wise affine operation used in the update step are special cases of a standard affine layer, which we review in Sec. 4.3.1.

### 4.3.1 Standard Affine Transformation

Suppose we apply an affine transformation to node $m$ at layer $l$ with state $x_t^{l,m}$

$$
x_t^{l+1,m} = W^l x_t^{l,m} + b^l, \tag{20}
$$

with weight matrix $W^l$ and bias $b^l$. The output moments are analytically tractable as

$$
\mathbb{E}[x_t^{l+1,m}] = W^l \mathbb{E}[x_t^{l,m}] + b^l, \qquad\qquad \mathrm{Cov}[x_t^{l+1,m}] = W^l \mathrm{Cov}[x_t^{l,m}](W^l)^T.
$$

The expected Jacobian of the affine transformation reads as $J_t^l = W^l$.

### 4.3.2 Node-Wise Affine Transformation

The node-wise affine transformation applies to each node $m$ at layer $l$ with state $x_t^{l,m}$ the same transformation simultaneously with weight matrix $W^l$ and bias $b^l$. The node-wise affine transformation can be interpreted

as a standard affine transformation acting on the set of all nodes $x_t^l$ as

$$
\begin{bmatrix} W^l x_t^{l,1} + b^l \\ W^l x_t^{l,2} + b^l \\ \vdots \\ W^l x_t^{l,M} + b^l \end{bmatrix} = \begin{bmatrix} W^l & 0 & \dots & 0 \\ 0 & W^l & \dots & 0 \\ \vdots & \vdots & \ddots & \vdots \\ 0 & 0 & \dots & W^l \end{bmatrix} \begin{bmatrix} x_t^{l,1} \\ x_t^{l,2} \\ \vdots \\ x_t^{l,M} \end{bmatrix} + \begin{bmatrix} b^l \\ b^l \\ \vdots \\ b^l \end{bmatrix} = \underbrace{(I_M \otimes W^l)}_{\hat{W}^l} x_t^l + \underbrace{(\mathbf{1}_M \otimes b^l)}_{\hat{b}^l}, \tag{21}
$$

where $I_M$ is the identity matrix with shape $M \times M$, $\mathbf{1}_M$ is a vector of ones with shape $M \times 1$, and $\otimes$ is the Kronecker product. The output moments of node-wise affine transformation are

$$
\mathbb{E}[x_t^{l+1}] = \hat{W}^l \mathbb{E}[x_t^l] + \hat{b}^l, \qquad\qquad \mathrm{Cov}[x_t^{l+1}] = \hat{W}^l \mathrm{Cov}[x_t^l](\hat{W}^l)^T.
$$

Similarly as for the standard affine transformation, the expected Jacobian of node-wise affine transformation is analytically available as $J_t^l = \hat{W}^l$.

### 4.3.3 Mean Aggregation

A commonly used aggregation operation is the mean aggregator, which calculates the message $x_t^{l,\mathcal{N}_m}$ to node $m$ at time step $t$ at layer $l$ as

$$
x_t^{l,\mathcal{N}_m} = \frac{1}{|\mathcal{N}_m|} \sum_{m' \in \mathcal{N}_m} x_t^{l,m'}. \tag{22}
$$

Let $x_t^{l,\mathcal{N}}$ be the set of all messages, i.e. $x_t^{l,\mathcal{N}} = \{x_t^{l,\mathcal{N}_m}\}_{m=1}^M$. The mean aggregation can be equivalently written as a linear transformation

$$
x_t^{l,\mathcal{N}} = \underbrace{(A \otimes I_{D_{x,l}})}_{\hat{A}} x_t^l. \tag{23}
$$

Above, $A \in \mathbb{R}^{M \times M}$ denotes the row normalized adjacency matrix, which summarizes the edge information $\mathcal{E}$ in matrix format and $I_{D_{x,l}}$ denotes the identity matrix with dimensionality $D_{x,l} \times D_{x,l}$. The Kronecker product expands the adjacency matrix accordingly to the $D_{x,l}$-dimensional node features. Hence, the mean aggregation corresponds to a linear transformation with a weight matrix consisting of $M \times M$ blocks, where each block is a diagonal matrix of shape $D_x \times D_x$. Its moments are analytically available as

$$
\mathbb{E}[x_t^{l,\mathcal{N}}] = \hat{A}\mathbb{E}[x_t^l], \qquad\qquad \mathrm{Cov}[x_t^{l,\mathcal{N}}] = \hat{A}\mathrm{Cov}[x_t^l]\hat{A}^T. \tag{24}
$$

The expected Jacobian is available as $J_t^l = \hat{A}$.

### 4.4 Sparse Covariance Approximation

For settings with a large number of agents $M$ or with a high-dimensional state $x_t^{l,m}$, the application of the BMM algorithm can become computationally expensive. In the following, we review the computational complexity of the BMM algorithm. We assume that the GNN model consists of a mean aggregation step followed by multiple node-wise affine transformations and nonlinearities, which is the same architecture that we employ later on in our experiments. The mean aggregation is done for each of the $D_x$ latent states independently, and we denote the maximum hidden layer width with $H$.

Computing the nonlinearities is cheap as the operation acts elementwise and their effect on the runtime can be neglected during this analysis. The other two operations (see Sec. 4.3.2 and Sec. 4.3.3) can be described by affine operations for which the weight matrices are heavily structured; the mean aggregation step corresponds to a weight matrix consisting of $M \times M$ diagonal blocks, the node-wise affine transformation to a block-diagonal weight matrix with $M$ blocks of shape $H \times H$. Propagation of the full covariance matrix through a neural network (forward cost) has the computational complexity of $\mathcal{O}(M^2 H^3 + M^2 H^2 D_x + M^2 H D_x^2 + M^3 D_x^2)$, where the first term is due to the cost of the $H \times H$-dimensional node-wise affine transformations in the hidden layers, the second and third term are due to the cost of the $H \times D$-dimensional node-wise affine transformation after the aggregation operation, and the fourth term is due to the aggregation operation.

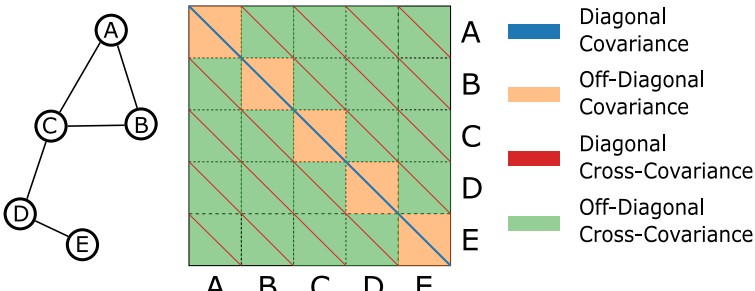

Figure 3: Groupings of the covariance matrix for graph structured data. Consider the example graph with $M = 5$ agents, which is depicted in the left panel. Each agent has dimensionality $D_x$. In the middle panel, we show the covariance matrix between all agents, which consists of $5 \times 5$ blocks, where each block has dimensionality $D_x \times D_x$.

The computational cost of the expected Jacobian is $\mathcal{O}(MH^3 + MH^2D_x + M^2D_x^2)$ and we give its derivation in the following. Let the expected Jacobian of the aggregation operation be $\mathbb{E}[J_t^1]$ and the product of the expected Jacobians of the subsequent neural net layers $\mathbb{E}[J_t^{net}] = \prod_{l=2}^L \mathbb{E}[J_t^l]$. The expected Jacobian of the neural net layers $\mathbb{E}[J_t^{net}]$ is block-diagonal with $M$ blocks of shape $D_x \times D_x$ and its computation takes $\mathcal{O}(MH^3 + MH^2D_x)$ time. Multiplying $\mathbb{E}[J_t^1]$ with $\mathbb{E}[J_t^{net}]$ results in a fully populated matrix where each entry can be computed by a single dot product, due to the structure of its factors, and its computation contributes with $\mathcal{O}(M^2D_x^2)$ to the runtime.

Consequently, the total cost is dominated by the forward cost, $\mathcal{O}(M^2H^3 + M^2H^2D_x + M^2HD_x^2 + M^3D_x^2)$,when using the full covariance matrix.[1] Since the computational cost quickly becomes intractable due to the cubic dependence with respect to the number of agents in the forward pass, we next propose different sparse approximations to the covariance matrix. Independent of the chosen approximation, the cost of the expected Jacobian remains unchanged, as it does not depend on the covariance matrix.

- **Full**: Model the full covariance matrix.
  Forward cost: $\mathcal{O}(M^2H^3 + M^2H^2D_x + M^2HD_x^2 + M^3D_x^2)$.
  Total cost: $\mathcal{O}(M^2H^3 + M^2H^2D_x + M^2HD_x^2 + M^3D_x^2)$.

- **Main Diagonal**: Keep the diagonal entries in the covariance blocks, which corresponds to the blue line in Fig. 3.
  Forward cost: $\mathcal{O}(MH^2 + MHD_x + M^2D_x)$.
  Total cost: $\mathcal{O}(MH^3 + MH^2D_x + M^2D_x^2)$.

- **Main Blocks**: Keep the block-diagonal blocks in the covariance matrix, which corresponds to the orange blocks and blue lines in Fig. 3.
  Forward cost: $\mathcal{O}(MH^3 + MH^2D_x + MHD_x^2 + M^2D_x^2)$.
  Total cost: $\mathcal{O}(MH^3 + MH^2D_x + MHD_x^2 + M^2D_x^2)$.

- **All Diagonals**: Structure the covariance matrix in blocks of shape $M \times M$ and keep the diagonal entries in each block, which corresponds to the blue and red lines in Fig. 3.
  Forward cost: $\mathcal{O}(M^2H^2 + M^2HD_x + M^3D_x)$.
  Total cost: $\mathcal{O}(M^2H^2 + M^2HD_x + M^3D_x + MH^3 + MH^2D_x + M^2D_x^2)$.

Note that setting the off-diagonal blocks to zero, as done in Main Blocks and Main Diagonal, corresponds to an independence assumption between the agents and leads to a runtime reduction from $O(M^3)$ to $O(M^2)$. In contrast, assuming a diagonal structure within each block, as performed in Main Diagonal and All Diagonals, corresponds to an independence assumption between the features and leads to runtime reduction from $O(H^3)$ to $O(H^2)$ in the forward pass.

---

[1]For reference, taking a Monte Carlo approach has the computational complexity $\mathcal{O}(SMH^2 + SMHD_x + SM^2D_x)$ where $S$ is the number of Monte Carlo samples.

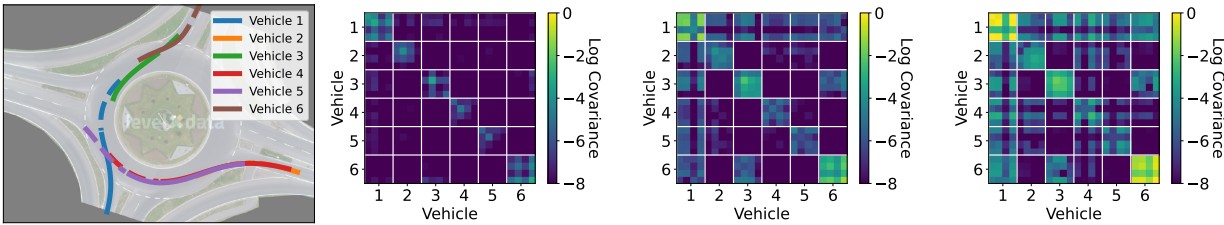

(a) Example Scene. (b) Covariance at 1 second. (c) Covariance at 3 seconds. (d) Covariance at 5 seconds.

Figure 4: We visualize an example scene from the rounD dataset (Krajewski et al., 2020) and the covariance matrix in the latent space at different time steps for the case of a unimodal initial distribution. In the left plot, dashed lines represent the observed history and solid lines represent the true future trajectories.

Finally, it is important to note that the covariance matrix only has the same structure as the graph after the first time step. Agents that are not connected via an edge can still have a non-zero cross-covariance at time step $t$, provided that they are connected by a path that is at most $t$ steps long. In our applications, this leads to non-sparse covariances after a few time steps, since the number of agents is small compared to the time horizon. We present the covariance matrix at three different time steps for an exemplary scene in Fig. 4. For a short prediction horizon of one second, the covariance matrix has an approximately diagonal shape. As the prediction horizon increases, the covariance matrix becomes more complex and is no longer dominated by its diagonal entries. We remark that the we could further increase the information spread across agents by using an architecture with multiple aggregation steps within the GNN module if required.

## 5 Experiments

We provide experiments on two challenging autonomous driving datasets. The first experiment (Sec. 5.1) conducts an ablation study, while the second experiment (Sec. 5.2) benchmarks our model against state-of-the-art methods. We provide a runtime analysis and a benchmark of our proposed sparse covariance approximations in Sec. 5.3. In Sec. 5.4, we analyse the generalization capabilities of our model by benchmarking it on novel unseen traffic environments.

We provide details about the training procedure in App. C, and the architecture for the embedding, transition, and emission functions in App. H. Accompanying code is available under `https://github.com/boschresearch/Deterministic-Graph-Deep-State-Space-Models`.

As evaluation metrics, we use the Root-Mean-Square Error (RMSE) and the predictive negative log-likelihood (NLL) at fixed time intervals (1s, 2s, 3s, 4s, 5s). For more details about the evaluation, we refer to App. D.

### 5.1 rounD

The rounD dataset (Krajewski et al., 2020) consists of vehicle trajectories recorded at different roundabouts in Germany. As the roundabouts involve many interactions among vehicles, we expect the predictive distributions to be multimodal and highly complex. We use the recordings from the roundabout in Neuweiler near Aachen for training and testing purposes. The dataset consists of 13,129 tracked objects recorded at 25 Hz in 22 sessions, amounting to a total recording time of 6.6 hours. We remove pedestrians, bicycles, as well as parked vehicles from the dataset, as their influence on the vehicle behaviour patterns in the roundabouts is negligible. After dataset curation, we are left with 12,715 tracked objects. We downsample the recordings by a factor of five and construct a dataset consisting of eight-second-long segments with 50 % overlap, resulting in 5405 snippets. We use the first three seconds as the track history, which is part of the context variable $\mathcal{I}$, and the following five seconds as the prediction horizon. The first 18 recording sessions, corresponding to 4,314 snippets, are used for training and validation. The final four recording sessions, corresponding to 1,091 snippets, are used for testing.

We build the connection graph by connecting vehicles with an Euclidean distance of less than 30 meters. The dataset has been recorded with drones and we keep the original global coordinate system.

### 5.1.1 Baselines

We compare our method with multiple baselines models. For each baseline, we remove one key assumption of our model. We cite papers that employ similar ideas as appropriate. We did not reimplement these works but performed an ablation study in which we replaced specific components of our model in the interest of a fair comparison. Furthermore, we compare against different training strategies as an alternative to maximizing the PLL and analyze the multimodality of our model.

*(i) Model*:

    *(i.i) GDSSM*: Our model as proposed in Sec. 4.

    *(i.ii) Non Recurrent GNN* (e.g. Casas et al. (2020); Herman et al. (2022)): This architecture receives the context variable $\mathcal{I}$, performs one round of message passing, and subsequently outputs one normal distribution for each of the next five seconds without using a recurrent architecture.

    *(i.iii) No Latent Noise* (e.g. Sanchez-Gonzalez et al. (2020)): We remove the noise from the latent dynamics [Eq. (2)] while keeping the emission model unchanged. The uncertainty can no longer be propagated forward in time as the emission model acts independently for each time point.

    *(i.iv) Linearity* (e.g. Li et al. (2020)): We remove all non-linearities from the latent dynamics. Note that we keep the non-linear emission model in order to map the dynamics into a latent space in which the system can be linearly approximated.

    *(i.v) No Interactions* (e.g. Krishnan et al. (2017)): Our model with a diagonal adjacency matrix, i.e. we remove all edges from the graphs. This model neglects interactions between traffic participants.

*(ii) Modes*:

    *(ii.i) V Modes*: We vary the number of mixture components $V$ in the GMM prior [Eq.(12)] in order to test the effect of multimodality.

    *(ii.ii) $\infty$ Modes*: This alternative does not follow an assumed density approach, i.e. the marginal latent distribution $p(x_t|\mathcal{I})$ is not approximated as a GMM with a bounded number of modes. Instead $p(x_t|\mathcal{I})$ is approximated by simulating a large number of trajectories, where each trajectory can follow a different mode (Brandt & Santa-Clara, 2002). We set the number of particles to 100.

*(iii) Objectives*:

    *(iii.i) PLL/Det.*: We train the model on the predictive log-likelihood [Eq.(15)] and use our deterministic approximations for GDSSM as proposed in Sec. 4.2 and Sec. 4.3.

    *(iii.ii) PLL/MC*: We train the model on the predictive log-likelihood [Eq.(15)] and take an assumed density approach by approximating the marginal distribution [Eq.(17)] as a GMM. The intractable integrals are solved via *Monte Carlo* (MC) integration. One forward pass through our model amounts approximately to the computational cost of 12 Monte Carlo simulations. For a fair comparison, we use during training 16 particles, which is more costly than training with our proposed moment propagation algorithm, and test with 100 particles. We use the same amount of particles for all MC based methods.

    *(iii.iii) ELBO/MC* (e.g. (Krishnan et al., 2017)): The model is trained by maximizing the *Evidence Lower Bound* (ELBO). We give a description of this loss function in App. E. The approximate posterior is a filtering distribution that models the latent state as a normal distribution. We propagate the latent state forward in time direction via Monte Carlo sampling.

*(iii.iv) MCO/MC* (e.g. (Maddison et al., 2017)): The model is trained by maximizing the *Monte Carlo Objective* (MCO), which combines particle filters with variational inference. We give a description of this loss function in App. E. The proposal distribution is a filtering distribution and we propagate the particles in time direction via Monte Carlo sampling.

### 5.1.2 Results

We provide benchmark results of all methods in Tab. 1. First, we compare our deterministic training and testing scheme (GDSSM PLL/Det.) against its Monte Carlo alternative (GDSSM PLL/MC). Though Monte Carlo based training is more costly than training with BMM, the Monte Carlo results are significantly outperformed by our method. Our results indicate that our deterministic approach leads to more effective approximations compared to Monte Carlo sampling despite the approximation error we obtain by our deterministic moment matching scheme. One potential explanation for our finding is that the Monte Carlo approaches suffer under a high variance since the latent space for multi-agent space grows linearly with the number of agents.

When changing the training objective from the PLL to ELBO/MC or MCO/MC, we observe that the performance is comparable to PLL/MC for the prediction horizon of 1 second. For longer prediction horizons, the performance of the models that are trained on ELBO/MC or MCO/MC degrade quicker compared to the model trained on PLL. We believe that this behavior can be explained by the mismatch between training and testing objectives. When training on MCO/MC or ELBO/MC, the model obtains feedback from the observations via the proposal distribution, while during testing it has to produce multi-step ahead predictions without receiving any feedback from the observations.

Next, we study if our method can capture multimodality by increasing the number of components in the GMM prior. It is worth noting that GMMs can approximate arbitrarily complex distributions when the number of components is chosen high enough. In our experiments, we observe that an increase of components significantly decreases the NLL, making the GMM prior a vital ingredient of our method and suggesting that the true predictive distribution is highly multi-modal. If the number of components is chosen too small, our model adjusts its uncertainty predictions accordingly. For example, we observe in Fig. 1 the uncertainty of the orange agent to increase as the vehicle is close to the exit of the roundabout. The high predictive uncertainty can be explained by two potential future outcomes: the orange vehicle can leave the roundabout or stay inside. Consequently, our model learns to compensate if the number of components is picked too low, which in turn allows us to trade accuracy for computational runtime. When using tailored implementations, one can easily scale up to a larger number of components, since their computations can be parallelized without any hurdles. We further note that the RMSE is less affected by the choice of the number of modes and stays within two standard errors. Since the RMSE value measures the error between the true value and the expected value of the predictive distribution, this result suggests that the expected value can already be modeled well by predictive distributions with very few modes (actually, one mode seems to suffice for this). We further provide the minRMSE values in App.F that decrease with increasing number of modes, indicating that the different modes correspond to a diverse set of plausible trajectories.

Next, we compare our model to a simpler alternative in which the dynamics are assumed to be linear, which allows calculating the moments of the transition model exactly and in closed form (Särkkä & Solin, 2019). In contrast, our approach, GDSSM, approximates the non-linear dynamics in a local linear way by using deterministic moment matching results. We find that our approach achieves lower RMSE and NLL, which can most likely be attributed to the higher modeling flexibility of our proposed model class.

Our ablation study further shows that discarding latent noise and removing interactions between traffic participants results in higher RMSE and NLL. Compared to modeling the dynamics in a non-recurrent manner, our model achieves a similar RMSE and outperforms its competitor in terms of NLL.

## 5.2 NGSIM

The *Next Generation Simulation* (NGSIM) dataset (Halkias & Colyar, 2007) consists of vehicle trajectories recorded at 10 Hz at two different highways, US-101 and I-80, in the United States. The dataset is commonly

Table 1: RounD results. We provide RMSE and NLL (mean ± standard error over 10 runs).

| | | Non Recur. GNN | GDSSM | GDSSM | GDSSM No Lat. Noise | GDSSM Linear | GDSSM No Interaction | GDSSM | GDSSM | GDSSM | GDSSM | GDSSM |
|---|---|---|---|---|---|---|---|---|---|---|---|---|
| | | 1 Mode PLL/Det. | ∞ Modes ELBO/MC | ∞ Modes MCO/MC | 1 Mode PLL/Det. | 1 Mode PLL/Det. | 1 Mode PLL/Det. | 1 Mode PLL/MC | 1 Mode PLL/Det. | 2 Modes PLL/Det. | 3 Modes PLL/Det. | 4 Modes PLL/Det. |
| RMSE | 1 s | $0.72 \pm 0.02$ | $1.53 \pm 0.04$ | $0.95 \pm 0.04$ | $1.22 \pm 0.08$ | $0.88 \pm 0.02$ | $0.91 \pm 0.03$ | $0.90 \pm 0.02$ | $0.79 \pm 0.02$ | $0.75 \pm 0.02$ | $0.74 \pm 0.03$ | $0.73 \pm 0.03$ |
| | 2 s | $1.90 \pm 0.02$ | $3.42 \pm 0.09$ | $2.43 \pm 0.09$ | $2.33 \pm 0.08$ | $2.12 \pm 0.03$ | $2.10 \pm 0.04$ | $2.09 \pm 0.02$ | $1.87 \pm 0.03$ | $1.84 \pm 0.03$ | $1.82 \pm 0.03$ | $1.80 \pm 0.04$ |
| | 3 s | $3.54 \pm 0.03$ | $5.99 \pm 0.14$ | $4.55 \pm 0.16$ | $3.88 \pm 0.07$ | $3.88 \pm 0.05$ | $3.69 \pm 0.06$ | $3.60 \pm 0.04$ | $3.36 \pm 0.03$ | $3.35 \pm 0.03$ | $3.35 \pm 0.03$ | $3.33 \pm 0.04$ |
| | 4 s | $5.26 \pm 0.04$ | $9.24 \pm 0.26$ | $7.62 \pm 0.27$ | $5.58 \pm 0.08$ | $6.13 \pm 0.09$ | $5.39 \pm 0.13$ | $5.37 \pm 0.05$ | $5.08 \pm 0.04$ | $5.15 \pm 0.09$ | $5.14 \pm 0.08$ | $5.06 \pm 0.04$ |
| | 5 s | $7.20 \pm 0.05$ | $11.59 \pm 0.41$ | $11.24 \pm 0.41$ | $7.65 \pm 0.10$ | $8.83 \pm 0.13$ | $7.56 \pm 0.23$ | $7.65 \pm 0.06$ | $7.24 \pm 0.05$ | $7.34 \pm 0.12$ | $7.35 \pm 0.19$ | $7.29 \pm 0.09$ |
| NLL | 1 s | $1.90 \pm 0.01$ | $2.62 \pm 0.04$ | $2.79 \pm 0.06$ | $2.96 \pm 0.08$ | $1.95 \pm 0.08$ | $1.67 \pm 0.07$ | $2.82 \pm 0.03$ | $1.48 \pm 0.05$ | $1.34 \pm 0.08$ | $1.36 \pm 0.04$ | $1.21 \pm 0.05$ |
| | 2 s | $3.25 \pm 0.02$ | $4.45 \pm 0.07$ | $4.21 \pm 0.03$ | $3.85 \pm 0.10$ | $3.93 \pm 0.13$ | $3.37 \pm 0.08$ | $4.11 \pm 0.02$ | $2.91 \pm 0.03$ | $2.93 \pm 0.07$ | $2.93 \pm 0.06$ | $2.79 \pm 0.09$ |
| | 3 s | $4.40 \pm 0.02$ | $5.64 \pm 0.09$ | $5.21 \pm 0.03$ | $5.05 \pm 0.13$ | $5.11 \pm 0.19$ | $4.34 \pm 0.08$ | $4.67 \pm 0.09$ | $3.87 \pm 0.02$ | $4.01 \pm 0.07$ | $3.77 \pm 0.10$ | $3.83 \pm 0.08$ |
| | 4 s | $5.13 \pm 0.02$ | $6.59 \pm 0.13$ | $6.01 \pm 0.03$ | $6.17 \pm 0.16$ | $6.01 \pm 0.22$ | $4.93 \pm 0.09$ | $5.01 \pm 0.07$ | $4.46 \pm 0.03$ | $4.52 \pm 0.11$ | $4.21 \pm 0.15$ | $4.34 \pm 0.18$ |
| | 5 s | $5.71 \pm 0.02$ | $6.98 \pm 0.14$ | $6.73 \pm 0.04$ | $7.24 \pm 0.20$ | $6.80 \pm 0.22$ | $5.51 \pm 0.10$ | $5.41 \pm 0.05$ | $5.05 \pm 0.04$ | $4.82 \pm 0.24$ | $4.59 \pm 0.15$ | $4.27 \pm 0.23$ |

used for benchmarking traffic forecasting methods and allows us to compare our method against prior art. We adopt the experimental setup of Deo & Trivedi (2018) and use both highway scenarios. As provided in the original publication, we employ a local coordinate system that is centered around an ego-vehicle.We split the scenarios into three 15 minute long time spans resembling mild, moderate, and congested traffic conditions and downsample each trajectory by a factor of two. The test set consists of a fourth of all trajectories randomly sampled from both locations. As in the rounD experiment, we split each trajectory into eight-second-long segments, where the first three seconds are used as the track history and the following five seconds as the prediction horizon.

Similarly as in Diehl et al. (2019); Lenz et al. (2017); Wheeler & Kochenderfer (2016), we introduce a connection graph based on the lane position of each vehicle. Each vehicle has at most six connections to other vehicles, which are the nearest vehicles in front/behind on the same/left/right lane. The vehicles on the outermost lanes have a maximum of four neighbours, as there are no neighbours to the left or right.

### 5.2.1 Baselines

We compare our approach with the following methods:

*(i) Constant Velocity (Mercat et al., 2019):* This method uses a linear state-space model together with a Kalman filter in order to make predictions. It does not take interactions between agents into account.

*(ii) Convolutional Social* (CS)-LSTM (Deo & Trivedi, 2018): Interactions between vehicles are modeled by introducing a grid and applying a convolutional layer on top. Dynamics are modeled by a deterministic LSTM, which predicts the mean and the variance of a normal distribution at each time step.

*(iv) Spatio-Temporal* (ST)-LSTM (Chen et al., 2020): Interactions are modeled with a spatio-temporal graph structure, i.e. a graph with a position and time depending component. Dynamics are modeled by a deterministic LSTM that predicts the mean and the variance of a normal distribution at each time step. To the best of our knowledge, this is the only GNN based method that also reports NLL.

*(iv) Multiple Futures Prediction* (MFP) (Tang & Salakhutdinov, 2019): A recurrent model with deterministic transition dynamics. Stochasticity is introduced via the initial state and noisy observations that are fed back into the dynamical model. Interactions are modeled by an attention module.

### 5.2.2 Results

We provide benchmark results of our proposed model in Tab. 2. Similar to the experiments on the rounD dataset in Sec. 5.1, we observe that (i) deterministic training and testing is more efficient than its Monte Carlo based alternative and (ii) increasing the number of modes improves the performance.

Next, we compare our method, using one component in the GMM prior only, with all other unimodal prediction methods. We can observe that our approach outperforms its competitors in terms of NLL. Furthermore, our method gives similar RMSE as other methods and is only outperformed by MFP. However, and in contrast to the other methods in the benchmark, MFP reports the best RMSE over 5 Monte Carlo samples, which makes it difficult to draw a fair conclusion.

We subsequently increase the number of modes for our model and for MFP. In terms of NLL, our method achieves superior results when it comes to long-term predictions (3s, 4s, 5s), while MFP performs better for short-term predictions (1s, 2s). Accurate long-term prediction of the agents in a driving scene is crucial for high-level autonomous driving, as an accurate environment model is a prerequisite for precise planning of driving controls. For instance, advanced driver-assistance systems usually use time horizons between 3s and 5s for driver warnings and emergency brakes, while autonomous cars aim for a time horizon for 5s or longer in order to ensure safe and comfortable rides (Philipp & Goehring, 2019). We provide minRMSE values for different number of modes as a measure of predictive diversity in App. F.

Table 2: NGSIM results. We provide RMSE and NLL averages and standard errors over 10 runs. For MFP, we report the best RMSE values over 5 Monte Carlo samples.

|  |  | Const. Vel. 1 Mode | CS-LSTM 1 Mode | ST-LSTM 1 Mode | MFP 1 Mode | MFP 4 Modes | GDSSM 1 Mode PLL/MC | GDSSM 1 Mode PLL/Det. | GDSSM 2 Modes PLL/Det. | GDSSM 3 Modes PLL/Det. | GDSSM 4 Modes PLL/Det. |
|---|---|---|---|---|---|---|---|---|---|---|---|
| RMSE | 1 s | 0.75 | 0.61 | 0.51 | 0.54 | 0.54 | $0.56 \pm 0.00$ | $0.53 \pm 0.01$ | $0.55 \pm 0.00$ | $0.55 \pm 0.01$ | $0.57 \pm 0.02$ |
|  | 2 s | 1.81 | 1.27 | 1.21 | 1.16 | 1.17 | $1.27 \pm 0.02$ | $1.18 \pm 0.01$ | $1.22 \pm 0.01$ | $1.23 \pm 0.02$ | $1.24 \pm 0.04$ |
|  | 3 s | 3.16 | 2.09 | 2.01 | 1.90 | 1.91 | $2.11 \pm 0.03$ | $1.98 \pm 0.02$ | $2.02 \pm 0.02$ | $2.05 \pm 0.03$ | $2.06 \pm 0.04$ |
|  | 4 s | 4.80 | 3.10 | 3.01 | 2.78 | 2.75 | $3.16 \pm 0.04$ | $2.99 \pm 0.03$ | $3.03 \pm 0.04$ | $3.06 \pm 0.05$ | $3.05 \pm 0.06$ |
|  | 5 s | 6.70 | 4.37 | 4.31 | 3.83 | 3.78 | $4.47 \pm 0.05$ | $4.29 \pm 0.04$ | $4.33 \pm 0.07$ | $4.33 \pm 0.07$ | $4.35 \pm 0.08$ |
| NLL | 1 s | 0.80 | 0.58 | 0.90 | 0.73 | -0.65 | $0.64 \pm 0.04$ | $0.19 \pm 0.02$ | $-0.12 \pm 0.02$ | $-0.15 \pm 0.03$ | $-0.16 \pm 0.03$ |
|  | 2 s | 2.30 | 2.14 | 2.41 | 2.33 | 1.19 | $2.10 \pm 0.10$ | $1.61 \pm 0.02$ | $1.37 \pm 0.02$ | $1.35 \pm 0.02$ | $1.32 \pm 0.02$ |
|  | 3 s | 3.21 | 3.03 | 3.25 | 3.17 | 2.28 | $2.92 \pm 0.11$ | $2.42 \pm 0.02$ | $2.23 \pm 0.02$ | $2.23 \pm 0.02$ | $2.19 \pm 0.02$ |
|  | 4 s | 3.89 | 3.68 | 3.61 | 3.77 | 3.06 | $3.54 \pm 0.10$ | $3.02 \pm 0.02$ | $2.88 \pm 0.02$ | $2.88 \pm 0.02$ | $2.82 \pm 0.02$ |
|  | 5 s | 4.44 | 4.22 | 4.36 | 4.26 | 3.69 | $4.04 \pm 0.10$ | $3.50 \pm 0.02$ | $3.42 \pm 0.02$ | $3.41 \pm 0.02$ | $3.36 \pm 0.02$ |

### 5.3 Covariance Approximations

We provide a detailed runtime analysis for different covariance approximations in Sec. 5.3.1 and study their impact on the predictive performance in Sec. 5.3.2.

#### 5.3.1 Runtime

We visualize the runtime of different covariance approximations, as well as the runtime of the Monte Carlo alternative in Fig. 5 as a function of input dimensionality and number of agents. We use the same NSDE architecture as in our experiments on the rounD and NGSIM dataset. For the Monte Carlo alternative, we visualize the runtime for 16 particles, as we use the same number of particles for training in Sec. 5.1 and 5.2.

We first confirm that propagation of the full covariance matrix is more costly than any of the proposed approximations. In fact, our sparse covariance approximations can reduce the runtime up to a factor of 100 for systems with a large number of agents and a high input dimensionality. As we derived in Sec. 4.4, when using the BMM algorithm with a full covariance matrix or the all diagonals approximation, the computational cost shows a cubic dependence on the number of agents. In contrast, the main diagonal approximation, the main blocks approximation, as well as MC based predictions have a quadratic dependence on the number of agents. This makes these two approximations an attractive alternative to MC based predictions, when systems with a high number of agents need to be modeled with a limited computational budget.

For systems with a moderate input dimensionality ($\approx 8$) and number of agents ($\approx 16$), all of our proposed approximations require only up to 5 ms in order to compute the distribution at the next time point, which corresponds to the cost of 5-10 Monte Carlo simulations.

#### 5.3.2 Benchmark

Next, we study the effect of different covariance approximations on the performance. We report the results for the case of a unimodal GDSSM. The results are depicted in Tab. 3. We report here our main findings.

First, modeling the full covariance matrix results in the best performance in terms of lowest RMSE and NLL. Second, the all diagonals covariance approximation performs the best among the covariance approximations.

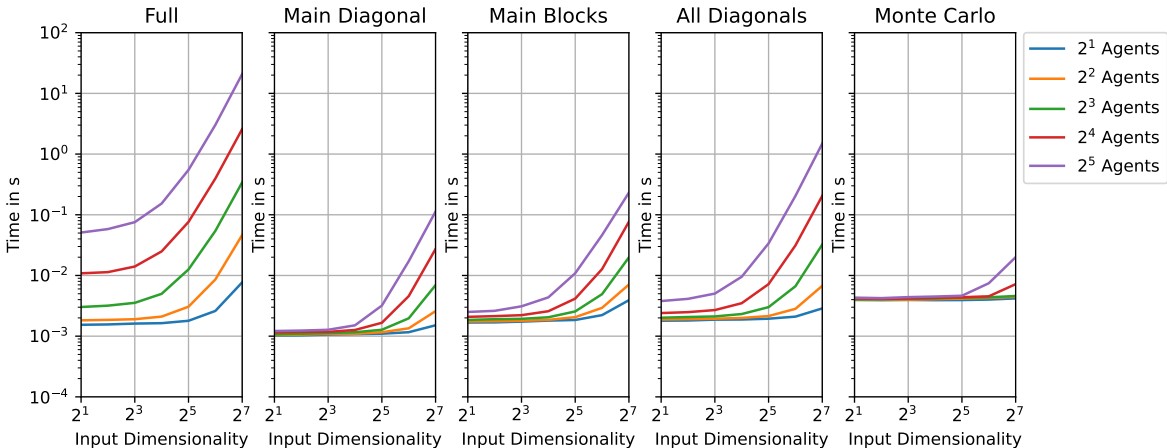

Figure 5: Wallclock time for output moment calculation for the latent dynamics using GNNs with three hidden layers of size 24 for different covariance approximations (from left to right). For each approximation, we plot the runtime as a function of the input dimensionality $D_x$ and number of agents $M$. The GNNs are initialized at random and we report the average runtime over 100 repetitions.

It achieves comparable RMSE as the full solution, but falls slightly behind in NLL when the system is highly interactive (see rounD dataset). In this setting, it outperforms the other two sparse approximations with respect to NLL. This behavior might be explained by the assumptions made in the covariance approximation: it is the only approximation that allows for correlations between agents as its structure only neglects dependencies between latent features.

Third, the differences in performances between the full solution and the different approximations with respect to RMSE lie between one and two standard errors. Modeling the main diagonal only can thus be sufficient for applications with low computational resources and a high demand in accuracy by accepting a slight loss in calibration. For these applications, the runtime can be reduced from $O(M^3)$ to $O(M^2)$.

Table 3: Test performance for different covariance approximations on the rounD and NGSIM dataset for the unimodal case. We provide average and standard error over 10 runs.

| | | | rounD | | | | NGSIM | | |
|---|---|---|---|---|---|---|---|---|---|
| | | Full | Main Diagonal | Main Blocks | All Diagonals | Full | Main Diagonal | Main Blocks | All Diagonals |
| RMSE | 1 s | $0.79 \pm 0.02$ | $0.82 \pm 0.02$ | $0.83 \pm 0.04$ | $0.78 \pm 0.02$ | $0.53 \pm 0.01$ | $0.54 \pm 0.01$ | $0.55 \pm 0.01$ | $0.53 \pm 0.01$ |
| | 2 s | $1.87 \pm 0.02$ | $1.88 \pm 0.02$ | $1.89 \pm 0.05$ | $1.87 \pm 0.02$ | $1.18 \pm 0.01$ | $1.18 \pm 0.02$ | $1.19 \pm 0.02$ | $1.19 \pm 0.02$ |
| | 3 s | $3.36 \pm 0.03$ | $3.40 \pm 0.02$ | $3.38 \pm 0.06$ | $3.34 \pm 0.02$ | $1.98 \pm 0.02$ | $1.99 \pm 0.03$ | $2.05 \pm 0.03$ | $1.99 \pm 0.02$ |
| | 4 s | $5.08 \pm 0.04$ | $5.07 \pm 0.04$ | $5.09 \pm 0.07$ | $5.05 \pm 0.03$ | $2.99 \pm 0.03$ | $2.98 \pm 0.04$ | $3.07 \pm 0.04$ | $2.97 \pm 0.03$ |
| | 5 s | $7.24 \pm 0.05$ | $7.25 \pm 0.06$ | $7.30 \pm 0.08$ | $7.25 \pm 0.05$ | $4.29 \pm 0.04$ | $4.34 \pm 0.05$ | $4.54 \pm 0.06$ | $4.32 \pm 0.04$ |
| NLL | 1 s | $1.48 \pm 0.05$ | $1.77 \pm 0.06$ | $1.79 \pm 0.05$ | $1.69 \pm 0.03$ | $0.19 \pm 0.02$ | $0.24 \pm 0.04$ | $0.22 \pm 0.03$ | $0.18 \pm 0.03$ |
| | 2 s | $2.91 \pm 0.03$ | $3.34 \pm 0.04$ | $3.35 \pm 0.06$ | $3.25 \pm 0.02$ | $1.61 \pm 0.02$ | $1.63 \pm 0.03$ | $1.64 \pm 0.03$ | $1.59 \pm 0.03$ |
| | 3 s | $3.87 \pm 0.02$ | $4.27 \pm 0.02$ | $4.28 \pm 0.05$ | $4.18 \pm 0.02$ | $2.42 \pm 0.02$ | $2.44 \pm 0.02$ | $2.46 \pm 0.02$ | $2.43 \pm 0.02$ |
| | 4 s | $4.46 \pm 0.03$ | $5.00 \pm 0.02$ | $5.01 \pm 0.05$ | $4.92 \pm 0.02$ | $3.02 \pm 0.02$ | $3.04 \pm 0.02$ | $3.06 \pm 0.04$ | $3.01 \pm 0.02$ |
| | 5 s | $5.05 \pm 0.04$ | $5.69 \pm 0.02$ | $5.68 \pm 0.06$ | $5.64 \pm 0.05$ | $3.50 \pm 0.02$ | $3.59 \pm 0.02$ | $3.62 \pm 0.03$ | $3.57 \pm 0.02$ |

## 5.4 Out-of-Distribution Testing

We analyze the generalization capabilities of our model by testing it on out-of-distribution data, e.g. traffic environments that have not been observed during training.

For this experiment, we reuse the rounD dataset (Krajewski et al., 2020) since it consists of recordings at three different roundabouts. Here, each roundabout corresponds to a separate traffic environment. We select one recording from each roundabout (Kackerstraße in Aachen, Thiergarten in Alsdorf and

Neuweiler near Aachen) and apply the same data curation steps as in Sec. 5.1. In order to generalize between different traffic environements, we change the experimental setup as follows:

- **Local coordinate system:** We transform the data into a local coordinate system, which is centered at the ego-vehicle and is oriented to the heading direction of the ego-vehicle.

- **Include map information to the context variable $\mathcal{I}$.** We extract a local map around the ego-vehicle, which spans a rectangle with a length of 74 meters and a width of 44 meters. Afterwards, we apply a binary masking to the map which divides the image into drivable and non-drivable areas. Our task is to jointly model all vehicles, which lie within this rectangle.

More details on the preprocessing can be found in App. G.

### 5.4.1 Results

Next, we analyze the generalization capabilities of our model by comparing the following strategies: (1) training on two traffic environments and testing on a third distinct traffic environment, (2) directly training the model on the test traffic environment and (3) training on all traffic environments. In order to enable a fair comparison, the data for each traffic environment is split into non-overlapping training and test sets.

We present the results for different traffic environments in Tab. 4. The first column in Tab. 4 describes the RMSE and NLL when we train our model on the traffic environments Thiergarten (T) and Neuweiler (N) and test it on the traffic environment Kackertstraße (K). The second column describes the predictive performance by using scenes from the same traffic environment (K) for training and testing. The third column describes the predictive performance by training on all three traffic environments (KNT) and testing on the traffic environment K. The remaining columns benchmark the generalization capabilities of our model on the traffic environments T and N and are set up in an analogous fashion.

In all experiments, we observe that the performance increases from (1) using different training environments for training and testing, to (2) using the same environment for training and testing and (3) using all environments for training. The difference in performance between (1) and (2) is moderate for the locations K and T demonstrating that our model is capable of generalizing to unseen traffic environments during testing. For the location N, the performance difference is increased which can be explained by studying the locations in more detail: N is a multi-lane roundabout with congested traffic, the roundabouts K and T are single-lane with moderate traffic. In consequence, the out-of-domain test on the traffic environment N is more challenging. We visualize exemplary predictions of our model GDSSM in Fig. 6.

Table 4: Test performance on different traffic environments on the rounD dataset using our method GDSSM (1 mode). We vary for each test traffic environment the training traffic environments. We provide averages and standard errors over 10 runs. Kackertstraße=K, Thiergarten=T, Neuweiler=N.

| Test | | K | | | T | | | N | |
|---|---|---|---|---|---|---|---|---|---|
| Train | TN | K | KTN | KN | T | KTN | KT | N | KTN |
| RMSE 1 s | 2.12 ± 0.09 | 1.88 ± 0.05 | 1.55 ± 0.04 | 1.42 ± 0.04 | 1.49 ± 0.05 | 1.21 ± 0.04 | 2.98± 0.12 | 2.02 ± 0.07 | 1.55 ± 0.04 |
| 2 s | 4.56 ± 0.13 | 3.73 ± 0.05 | 3.34 ± 0.09 | 2.82 ± 0.05 | 2.53 ± 0.09 | 2.27 ± 0.04 | 5.82± 0.19 | 3.38 ± 0.06 | 3.02 ± 0.09 |
| 3 s | 7.51 ± 0.24 | 6.05 ± 0.17 | 5.62 ± 0.20 | 4.42 ± 0.12 | 3.67 ± 0.12 | 3.35 ± 0.07 | 8.96± 0.24 | 4.89 ± 0.13 | 4.91 ± 0.18 |
| 4 s | 10.57 ± 0.34 | 8.65 ± 0.26 | 7.99 ± 0.17 | 6.75 ± 0.18 | 5.33 ± 0.15 | 5.00 ± 0.16 | 12.44± 0.33 | 6.81 ± 0.14 | 6.91 ± 0.18 |
| 5 s | 13.00 ± 0.31 | 12.02 ± 0.43 | 10.73 ± 0.27 | 9.97 ± 0.40 | 7.59 ± 0.29 | 7.41 ± 0.27 | 14.86± 0.42 | 8.77 ± 0.19 | 8.69 ± 0.31 |
| NLL 1 s | 4.11 ± 0.07 | 4.02 ± 0.04 | 3.13 ± 0.06 | 3.35 ± 0.06 | 3.16 ± 0.06 | 2.88 ± 0.06 | 4.75± 0.07 | 3.84 ± 0.04 | 3.21 ± 0.08 |
| 2 s | 5.36 ± 0.10 | 4.84 ± 0.03 | 4.66 ± 0.09 | 4.22 ± 0.04 | 3.99 ± 0.07 | 3.65 ± 0.04 | 5.82± 0.09 | 4.60 ± 0.05 | 4.42 ± 0.09 |
| 3 s | 6.55 ± 0.17 | 5.97 ± 0.07 | 6.42 ± 0.17 | 5.49 ± 0.10 | 4.98 ± 0.11 | 4.50 ± 0.05 | 7.32± 0.17 | 5.55 ± 0.04 | 5.68 ± 0.09 |
| 4 s | 7.71 ± 0.20 | 7.07 ± 0.14 | 7.57 ± 0.21 | 7.20 ± 0.09 | 6.38 ± 0.19 | 5.89 ± 0.13 | 8.65± 0.18 | 6.79 ± 0.11 | 6.71 ± 0.08 |
| 5 s | 8.63 ± 0.18 | 8.14 ± 0.18 | 8.08 ± 0.15 | 8.54 ± 0.12 | 7.45 ± 0.21 | 6.84 ± 0.11 | 9.11± 0.18 | 7.54 ± 0.22 | 7.04 ± 0.14 |

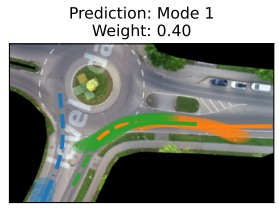 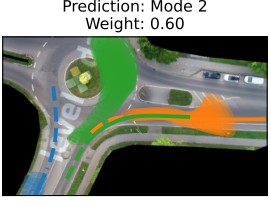 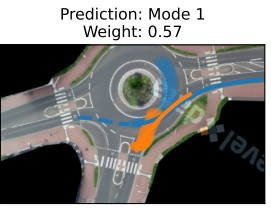 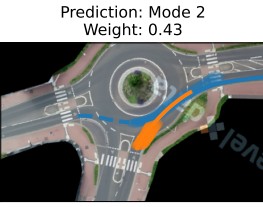

(a) Kackertstraße (K)               (b) Thiergarten (T)

Figure 6: Predictions of our model GDSSM (2 modes) on out-of-domain traffic environments, i.e. the training and testing traffic environments are distinct. Given the history of each traffic participant (dashed lines), we visualize the predicted 95% confidence interval. Solid lines represent the true future trajectory.

## 6  Conclusion

In this work, we have proposed GDSSMs in which the latent dynamics of the agents are coupled via GNNs in order to capture interactions among multiple agents. We derived moment matching rules for GNN layers that allow for deterministic inference and introduced a GMM prior over the initial latent states in order to allow for multimodal predictions. Both together lead to an efficient and stable algorithm that is able to produce complex and nonlinear predictive distributions. We confirmed that our novel method shows strong empirical performance on two challenging autonomous driving datasets. Finally, we proposed sparse approximations to the covariance matrix considering the computational limits of real-world vehicle control units. Depending on the required calibration, our approximations can lead to a significant reduction of the runtime without impeding accuracy.

In future work, we seek to increase the robustness of our proposed model towards novel and unseen traffic scenarios. One way could be to incorporate epistemic uncertainty into our model formulation by placing a prior over the weights of the GNN. To achieve this, we could combine our model with recent advances in variational inference in order to find an approximation to the intractable weight posterior. Another research direction of interest is modeling of irregular and partially observed dynamical systems. Prior work uses continuous time encoder networks as well as a continuous time transition model in latent space (Rubanova et al., 2019; Brouwer et al., 2019). Following this vein of work, an extension of our model towards continuous time networks seems a promising direction for modeling interactive systems.

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

## A   Proof of Theorem 1

**Theorem 1.** *The marginal distribution $p(y_t|\mathcal{I})$ is analytically computed as*

$$p(y_t|\mathcal{I}) = \sum_{v=1}^{V} \pi_v(\mathcal{I})\mathcal{N}(y_t|a_{t,v}(\mathcal{I}), B_{t,v}(\mathcal{I})),$$

*for a GDSSM with the below generative model*

$$
\begin{aligned}
v &\sim Cat([\pi_1(\mathcal{I}), \dots, \pi_V(\mathcal{I})]), \\
x_0 &\sim \mathcal{N}(\mu_{0,v}(\mathcal{I}), diag(\Sigma_{0,v}(\mathcal{I}))), \\
x_t &\sim \mathcal{N}\left(x_t|x_{t-1} + f(t, v, \mathcal{I})x_{t-1}, diag\left(L(t, v, \mathcal{I})\right)\right), & t &= 1, \dots, T \\
y_t &\sim \mathcal{N}\left(y_t|g(t, v, \mathcal{I})x_t, diag\left(\Gamma(t, v, \mathcal{I})\right)\right), & t &= 1, \dots, T
\end{aligned}
$$

*where $f(t, v, \mathcal{I}), L(t, v, \mathcal{I}), g(t, v, \mathcal{I}), \Gamma(t, v, \mathcal{I})$ are time $t$, component $v$, and context $\mathcal{I}$ depending matrices with appropriate dimensionality.*

*Proof.* The proof is straightforward as the output moment of the transition and emission function are analytically available

$$
\begin{aligned}
p(y_t|\mathcal{I}) &= \int p(y_t|v, x_t, \mathcal{I})p(x_t|v, x_0, \mathcal{I})p(v, x_0|\mathcal{I})dv dx_0 dx_t \\
&= \sum_{v=1}^{V} \pi_v(\mathcal{I}) \int p(y_t|v, x_t, \mathcal{I})p(x_t|v, x_0, \mathcal{I})\mathcal{N}(x_0|\mu_{0,v}(\mathcal{I}), \operatorname{diag}(\Sigma_{0,v}(\mathcal{I})))dx_0 dx_t \\
&= \sum_{v=1}^{V} \int p(y_t|v, x_t, \mathcal{I})\mathcal{N}(x_t|\mu_{t,v}(\mathcal{I}), \Sigma_{t,v}(\mathcal{I}))dx_t \\
&= \sum_{v=1}^{V} \pi_v(\mathcal{I})\mathcal{N}(y_t|a_{t,v}(\mathcal{I}), B_{t,v}(\mathcal{I})).
\end{aligned}
$$

The moments at time step $t$ of a linear time depending dynamical system are available as (Särkkä, 2013)

$$
\begin{aligned}
\mu_{t,v}(\mathcal{I}) &= \prod_{t'=1}^{t} \left(I + f(t', v, \mathcal{I})\right) \mu_{0,v}(\mathcal{I}), \\
\Sigma_{t,v}(\mathcal{I}) &= \left[\prod_{t'=1}^{t} (I + f(t', v, \mathcal{I}))\right] \operatorname{diag}(\Sigma_{0,v}(\mathcal{I})) \left[\prod_{t'=1}^{t} (I + f(t', v, \mathcal{I}))^T\right] \\
&\quad + \sum_{t'=1}^{t-1} \left[\prod_{t''=t'+1}^{t} (I + f(t'', v, \mathcal{I}))\right] \operatorname{diag}(L(t', v, \mathcal{I})) \left[\prod_{t''=t'+1}^{t} (I + f(t'', v, \mathcal{I}))^T\right] \\
&\quad + \operatorname{diag}(L(t, v, \mathcal{I}))
\end{aligned}
$$

We obtain the same expression via the BMM algorithm, which is easy to prove by inserting the locally linear system into Eq. 6. Finally, mean $a_{t,v}(\mathcal{I})$ and covariance $B_{t,v}(\mathcal{I})$ of the output at time step $t$ are available as (Särkkä, 2013)

$$
\begin{aligned}
a_{t,v}(\mathcal{I}) &= \mathbb{E}[g(t, v, \mathcal{I})x_t] = g(t, v, \mathcal{I})\mu_{t,v}(\mathcal{I}), \\
B_{t,v}(\mathcal{I}) &= \operatorname{Cov}[g(t, v, \mathcal{I})x_t] + \operatorname{diag}\left(\mathbb{E}[\Gamma(t)]\right) = g(t, v, \mathcal{I})\Sigma_{t,v}(\mathcal{I})g(t, v, \mathcal{I})^T + \operatorname{diag}\left(\Gamma(t, v, \mathcal{I})\right).
\end{aligned}
$$

$\square$

# B  Output Moments for the ReLU Activation Function

Suppose, we apply the ReLU activation function at layer $l$ and time step $t$ to the input $x_t^l$

$$x_t^{l+1} = \max(0, x_t^l).$$

**Output Moments** We can approximate the output moments of $x_t^{l+1}$ as a function of the input moments (Wu et al., 2019)

$$\mathbb{E}[\max(0, x_t^l)] \approx \sqrt{\text{diag}(\text{Cov}[x_t^l])} \ \text{SR}\left(\mathbb{E}[x_t^l]/\sqrt{\text{diag}(\text{Cov}[x_t^l])}\right),$$

$$\text{Cov}[\max(0, x_t^l)] \approx \sqrt{\text{diag}(\text{Cov}[x_t^l])\text{diag}(\text{Cov}[x_t^l])^T} \ F\left(\mathbb{E}[x_t^l], \text{Cov}[x_t^l]\right),$$

where $\text{SR}(x) = \phi(x) + x\Phi(x)$ with $\phi$ and $\Phi$ representing the PDF and CDF of a standard Gaussian variable. The function $F(\mathbb{E}[x_t^l], \text{Cov}[x_t^l])$ is a shorthand notation for

$$F(\mathbb{E}[x_t^l], \text{Cov}[x_t^l]) = A\left(\mathbb{E}[x_t^l], \text{Cov}[x_t^l]\right) + \exp\left[-Q\left(\mathbb{E}[x_t^l], \text{Cov}[x_t^l]\right)\right].$$

The function $A(\mathbb{E}[x_t^l], \text{Cov}[x_t^l])$ can be estimated as

$$A(\mathbb{E}[x_t^l], \text{Cov}[x_t^l])) = \text{SR}(\epsilon_k^l)\text{SR}(\epsilon_k^l)^T + \rho_k^l \Phi(\epsilon_k^l)\Phi(\epsilon_k^l)^T,$$

where $\rho_k^l = \text{Cov}[x_t^l]/\left(\sqrt{\text{diag}(\text{Cov}[x_t^l])}\sqrt{\text{diag}(\text{Cov}[x_t^l])^T}\right)$ is a dimensionless matrix and $\epsilon_k^l = \mathbb{E}[x_t^l]/\sqrt{\text{diag}(\text{Cov}[x_t^l])}$ is a dimensionless vector.

The $i, j$-th element of $Q(\mathbb{E}[x_t^l], \text{Cov}[x_t^l])$ can be estimated as:

$$Q\left(\mathbb{E}[x_t^l], \text{Cov}[x_t^l]\right)_{i,j} = \frac{\rho_{k_{i,j}}^l}{2g_{k_{i,j}}^l(1 + \bar{\rho}_{k_{i,j}})}\left((\epsilon_{k_i}^l)^2 + (\epsilon_{k_j}^l)^2\right) - \frac{\arcsin(\rho_{k_{i,j}}^l) - \rho_{k_{i,j}}^l}{\rho_{k_{i,j}}g_{k_{i,j}}^l}\epsilon_{k_i}^l\epsilon_{k_j}^l - \log\left(\frac{g_{k_{i,j}}^l}{2\pi}\right),$$

where $g_k^l = \arcsin(\rho_k^l) + \rho_k^l \oslash \left(1 + \sqrt{1 - \rho_k^l \odot \rho_k^l}\right)$. The operator $\oslash$ denotes the elementwise disivision and $\odot$ denotes the elementwise multiplication.

**Expected Jacobian** Since activation functions are applied element-wise, off-diagonal entries of the expected gradient are zero. The diagonal of the Jacobian of the ReLU function is the expected Heaviside step function Wu et al. (2019)

$$\text{diag}\left(\mathbb{E}\left[\nabla_{x_t^l}\max(0, x_t^l)\right]\right) \approx \Phi\left(\mathbb{E}[x_t^l]/\sqrt{\text{diag}(\text{Cov}[x_t^l]))}\right).$$

For a more detailed derivation, we refer to the work of Wu et al. (2019).

# C  Training Details

We train all models with the ADAM optimizer and stochastic mini-batches. We use a batch size of 4 and a learning rate of 0.0001. In order to accelerate training of multi-modal GDSSMs we initialize the transition and observation neural nets with the pretrained versions of the uni-modal GDSSMs.

## C.1  rounD

We train deterministic models (GDSSM Det. and Non Recur. GNN) for 50k weight updates. For the MC based models (GDSSM MC) we use 100k weight updates. The dataset contains 4,314 training and 1,091 testing snippets. We do not use a separate validation dataset as we observed no overfitting.

### C.2 NGSIM

We train all models for 1000k updates on the NGSIM dataset. The dataset contains 5,922k/860k/1,505k train/validation/test snippets. We validate the models on 1k random minibatches from the validation dataset after every 10k weight updates.

### C.3 Out-of-Distribution Testing

We train all models for 50k weight updates. The dataset consists of three sub-datasets with 254/ 251/ 137 snippets, where 80% of each sub-dataset are used for training and the remaining 20% for testing. Due to the small size of the sub-datasets we observed overfitting. We address this by using the last 20% of each training sub-dataset for validation. We validate the model after every 1k weight updates.

## D  Evaluation Metrics

In the following, we give a brief overview on different evaluation metrics. We provide the negative log-likelihood (NLL) and the Root-Mean-Square Error (RMSE) in the main of the paper, and minRMSE in App. F.

**Root-Mean-Square Error**  The (root) mean-square error at time point $t$ is defined as

$$\text{MSE}_{\text{Base}}(t) = \frac{1}{N} \sum_{n}^{N} \sum_{m}^{M(n)} \frac{(y_{t,n}^m - \hat{y}_{t,n}^m)^T (y_{t,n}^m - \hat{y}_{t,n}^m)}{M(n)}, \quad \text{RMSE}_{\text{Base}}(t) = \sqrt{\text{MSE}_{\text{Base}}(t)}, \tag{25}$$

where $N$ is the number of snippets in the dataset, $M(n)$ is the number of agents in the $n$-th snippet, $y_{t,n}^m$ is the true location of the $m$-th agent at time point $t$ of the $n$-th snippet, and $\hat{y}_{t,n}^m$ the corresponding predicted location.

Unfortunately, it is not straight-forward to extend RMSE$_{\text{base}}$ to probabilistic forecasts. To the best of our knowledge, none of the existing variants is a strictly proper scoring rule (Gneiting & Raftery, 2007) where the latter is optimal if and only if the predictive distribution matches the true distribution. An example of a strictly proper scoring rule that we also provide in the paper is the predictive NLL:

$$\text{NLL}(t) = -\frac{1}{N} \sum_{n}^{N} \sum_{m}^{M(n)} \frac{\log p(y_{t,n}^m | \mathcal{I})}{M(n)}. \tag{26}$$

For deterministic GDSSMs, we approximate $p(y_{t,n}^m | \mathcal{I})$ directly with our method that we introduced in Sec. 4.2. For GDSSMs that do not follow an assumed density approach, we approximate $p(y_{t,n}^m | \mathcal{I})$ via Monte Carlo integration.

In the following, we discuss different RMSE variants in more detail.

**Bayes Predictor:**  A common evaluation metric used in the ML community (e.g. Gal & Ghahramani (2016)) is to compute

$$\text{RMSE}(t) = \sqrt{\frac{1}{N} \sum_{n}^{N} \sum_{m}^{M(n)} \frac{(y_{t,n}^m - \mathbb{E}[\hat{y}_{t,n}^m])^T (y_{t,n}^m - \mathbb{E}[\hat{y}_{t,n}^m])}{M(n)}}, \tag{27}$$

where we assume that the probabilistic predictor will perform model averaging before making its final prediction. This score only evaluates the goodness of the first moment and does not take any information of higher moments into account. We provide its values in the main part of the paper.

**Gibbs Predictor:**  An alternative way for extending deterministic loss functions to the probabilistic scenario can be found in the PAC-Bayes setting (e.g. Germain et al. (2016)) by considering the average loss in

the risk function. In our case, this would lead to

$$\text{RMSE}_{\text{Gibbs}}(t) = \sqrt{\mathbb{E}\left[\frac{1}{N}\sum_n^N \sum_m^{M(n)} \frac{(y_{t,n}^m - \hat{y}_{t,n}^m)^T (y_{t,n}^m - \hat{y}_{t,n}^m)}{M(n)}\right]}, \tag{28}$$

where we take expectation with respect to the predictions $\hat{y}_{t,n}^m$. However, we discourage to take this formulation as evaluation metric since it penalizes any form of variance in the predictor and favors a Dirac distribution around the expected value of the true distribution. Note that this result can be tightly linked to the bias-variance trade-off in statistical learning theory (e.g. Hastie et al. (2009)).

**Min Predictor:** A popular alternative in the traffic forecasting literature (e.g. Tang & Salakhutdinov (2019)) is to compute the minimal error over a set of $S$ potential predictions

$$\text{minRMSE}(t) = \sqrt{\frac{1}{N}\sum_n^N \sum_m^{M(n)} \min_{s=1,\ldots,S} \frac{(y_{t,n}^m - \hat{y}_{t,n,s}^m)^T (y_{t,n}^m - \hat{y}_{t,n,s}^m)}{M(n)}}, \tag{29}$$

where $(\hat{y}_{t,n,s}^m)_{s=1}^S$ is the set of $S$ potential predictions. This score favors models that produce a set of diverse predictions where at least one candidate is close to the true outcome but ignores the goodness of the remaining $S-1$ predictors. While it gives some useful information about the model capacity, it does not give a complete representation of the test-time performance when the best mode is unknown. We provide its values in App. F.

## E    Alternative Parameter Inference Methods

One commonly used inference method in the context of machine learning circumvents maximizing the log-likelihood and instead maximizes the *Evidence Lower Bound* (ELBO)

$$\text{ELBO}(y_1,\ldots,y_T|\mathcal{I}) = \mathbb{E}_{q(x_0,\ldots,x_T)}\left[\log \frac{p(y_1,\ldots,y_T,x_0,\ldots,x_T|\mathcal{I})}{q(x_0,\ldots,x_T)}\right], \tag{30}$$

that involves learning an approximation $q(x_0,\ldots,x_T)$ to the intractable smoothing distribution $p(x_0,\ldots,x_T|y_1,\ldots,y_T,\mathcal{I})$ (Krishnan et al., 2017).

A tighter bound to the log-likelihood $\log p(y_1,\ldots,y_T|\mathcal{I})$ can be obtained by calculating the importance weighted log-likelihood, which we refer to as the *Monte Carlo Objective* (MCO) (Maddison et al., 2017; Burda et al., 2016)

$$\text{MCO}(y_1,\ldots,y_T|\mathcal{I}) = \mathbb{E}_{q(x_0,\ldots,x_T)}\left[\log \frac{1}{K}\sum_{k=1}^K \frac{p(y_1,\ldots,y_T,x_{0,k},\ldots,x_{T,k}|\mathcal{I})}{q(x_{0,k}\ldots,x_{T,k})}\right], \tag{31}$$

where $K$ is the number of Monte Carlo samples and $x_{t,k}$ is the $k$-th sample at time step $t$. For state-space models, recent work combined the MCO with particle filters (Naesseth et al., 2018; Maddison et al., 2017; Le et al., 2018).

## F    Extended Results

We provide minRMSE values for the rounD and NGSIM dataset. We set the proposal predictions to the mean values of each mode and the min operator selects for each agent the proposal that produces the lowest error per snippet over the complete prediction horizon. This score favors models in which the proposal predictions are diverse as long as at least one candidate is close to the true outcome. As shown in Tab. 5, the minRMSE decreases as we increase the number of modes, which indicates that the different modes correspond to a diverse set of plausible trajectories.

Table 5: minRMSE values as a function of the number of modes on the rounD and NGSIM dataset. We provide average and standard error over 10 runs. For RMSE and NLL results, please refer to Tab. 1 and Tab. 2.

| | | rounD | | | | NGSIM | | | |
|---|---|---|---|---|---|---|---|---|---|
| | | 1 Mode | 2 Modes | 3 Modes | 4 Modes | 1 Mode | 2 Modes | 3 Modes | 4 Modes |
| minRMSE | 1 s | $0.79 \pm 0.02$ | $0.75 \pm 0.04$ | $0.76 \pm 0.03$ | $0.69 \pm 0.02$ | $0.53 \pm 0.01$ | $0.46 \pm 0.01$ | $0.35 \pm 0.01$ | $0.32 \pm 0.01$ |
| | 2 s | $1.87 \pm 0.02$ | $1.85 \pm 0.06$ | $1.83 \pm 0.07$ | $1.68 \pm 0.05$ | $1.18 \pm 0.01$ | $1.05 \pm 0.01$ | $0.80 \pm 0.01$ | $0.71 \pm 0.01$ |
| | 3 s | $3.36 \pm 0.03$ | $3.46 \pm 0.11$ | $3.05 \pm 0.16$ | $2.75 \pm 0.12$ | $1.98 \pm 0.02$ | $1.73 \pm 0.02$ | $1.33 \pm 0.02$ | $1.18 \pm 0.03$ |
| | 4 s | $5.08 \pm 0.04$ | $5.07 \pm 0.13$ | $4.55 \pm 0.28$ | $4.04 \pm 0.45$ | $2.99 \pm 0.03$ | $2.60 \pm 0.04$ | $2.02 \pm 0.04$ | $1.75 \pm 0.04$ |
| | 5 s | $7.24 \pm 0.05$ | $6.29 \pm 0.23$ | $5.95 \pm 0.47$ | $4.43 \pm 0.53$ | $4.29 \pm 0.04$ | $3.69 \pm 0.06$ | $2.88 \pm 0.05$ | $2.46 \pm 0.05$ |

## G   Experimental Setup for Out-of-Distribution Testing

### G.1   Dataset Construction

We construct a dataset, which consists of three different traffic environments. We select one recording from the roundabout in Kackertstraße (K) in Aachen, one recording from the roundabout in Thiergarten (T) in Alsdorf, and one recording from the roundabout in Neuweiler (N) near Aachen. We use the same preprocessing procedure as in Sec. 5.1. We remove pedestrians, bicycles, and parked vehicles from each traffic environment. There remain 319/ 264/ 389 tracked objects over a time span of 0.3/ 0.3/ 0.15 hours at the traffic environments K/ T/ N. We downsample the recordings by a factor of 5 and then construct for each traffic environment a dataset which consists of 8 s long snippets with 50% overlap. The first three seconds are used as the track history and the following five seconds as the prediction horizon. We obtain 254/ 251/ 137 snippets for the traffic environments K/ T/ N. For each traffic environment we use the first 80% snippets for training and the remaining 20% snippets for testing.

### G.2   Map Processing

The map processing follows existing work in the domain of traffic forecasting Bansal et al. (2018); Herman et al. (2022). Given an ego-vehicle and its history, we first calculate the heading of the vehicle. The heading is calculated as the average heading direction over the last 0.2 observed seconds. The position and the heading direction jointly define the *Region-of-Interest* (ROI) on the map. The ROI is a rectangle with a length of 74 meters and a width of 44 meters, which is centered at the ego-vehicle and oriented according to the heading of the ego-vehicle. We further convert the RGB-image into a binary road image. We visualize the processed map information in Fig. 7b

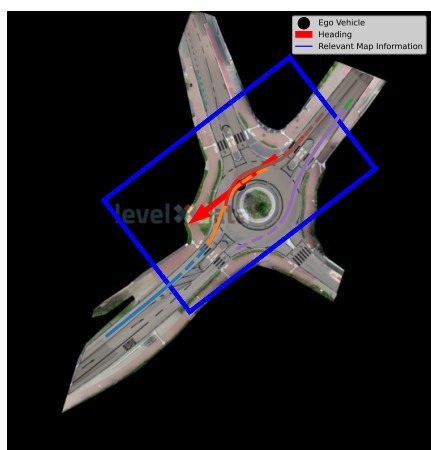
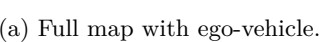

(a) Full map with ego-vehicle.

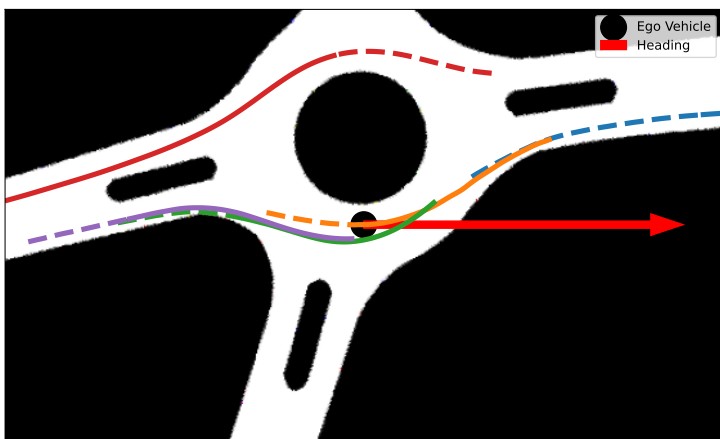

(b) Rotated, cropped, and masked map information.

Figure 7: Processing of map information.

## H  Network Architectures

### H.1  Network Architectures without World Models

Below we present the neural network architectures for the experiments in Sec. 5.1, 5.2, 5.3.

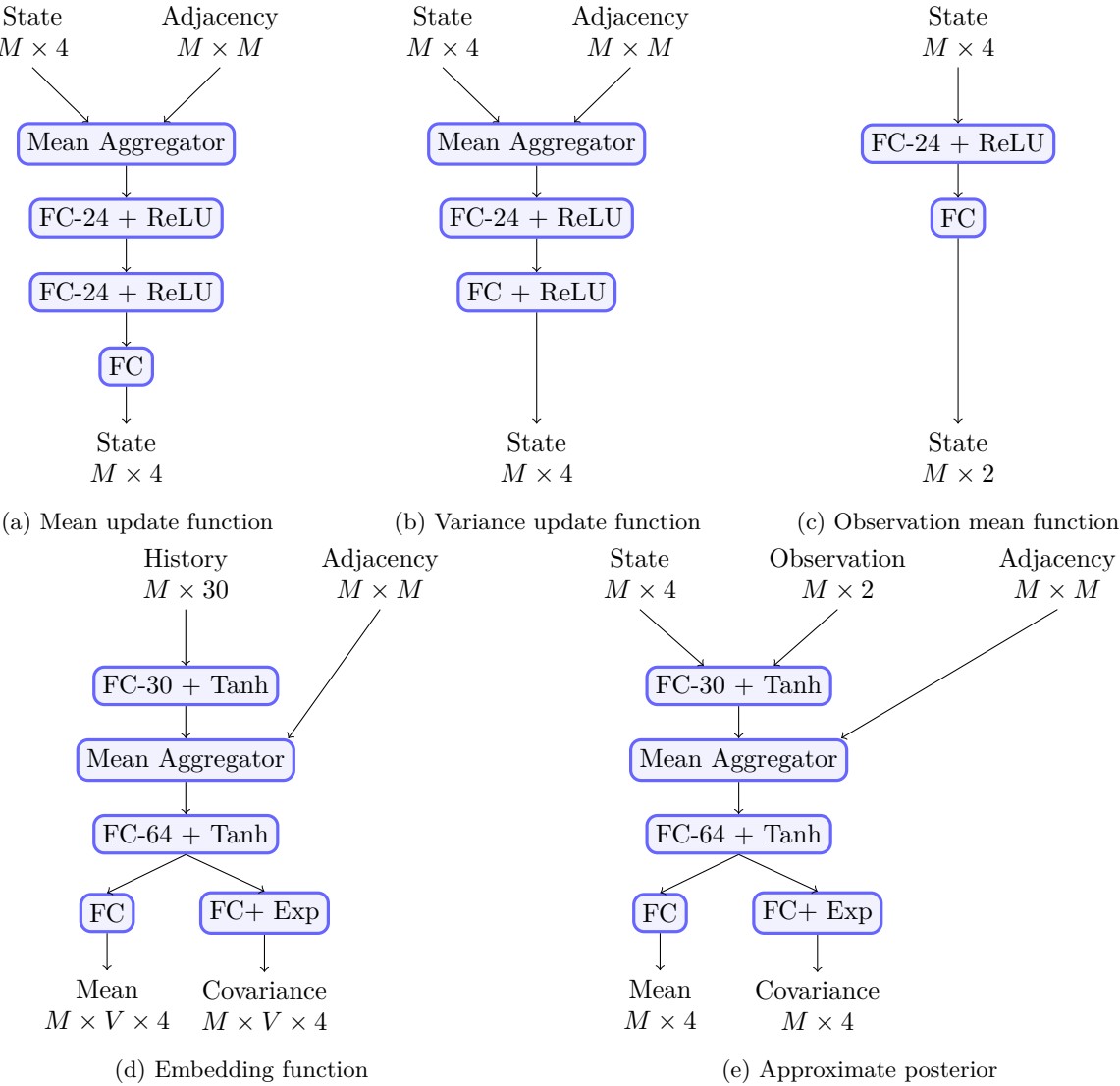

Figure 8: Architectures without map information. For fully connected layers we give the number of output neurons.

The embedding function receives the history of all $M$ agents. This history is 3 seconds long with a time step of 0.2 seconds and consists of two dimensional coordinates. After flattening, the input is a vector of size 30. The output of the embedding function is a GMM with $V$ mixture components. We use the mean aggregator in the embedding, mean and variance update function. The mean aggregator calculates the message to agent $m$ accordingly to Eq. 22. After the message $x_t^{\mathcal{N}_m}$ is calculated, it is concatenated with the state $x_t^m$. Mean and variance update functions are neural networks, which conduct at each prediction step one round of message passing and then calculate the output. Our emissions model uses a neural network for the mean function $g(x_t)$ and a constant vector for $Q(x_t)$. The emission model maps the state of each agent back to the observed space and does not depend on interactions. The approximate posterior is used

in Sec. 5.1 for the baselines that are trained on the ELBO or MCO. In contrast maximizing the PLL does not necessitate an approximate posterior. The approximate posterior network takes the previous state and the current observation in order to approximate the latent distribution at the current time step.

## H.2 Network Architectures with Map Information

These neural net architectures have been used in our experiments in Sec. 5.4. It closely follows the architectures in Sec. H.1. We add an additional neural net, which encodes the masked map of size 500x300 into a flattened 256-dimensional vector. This map embedding is used as an additional input to the embedding function.

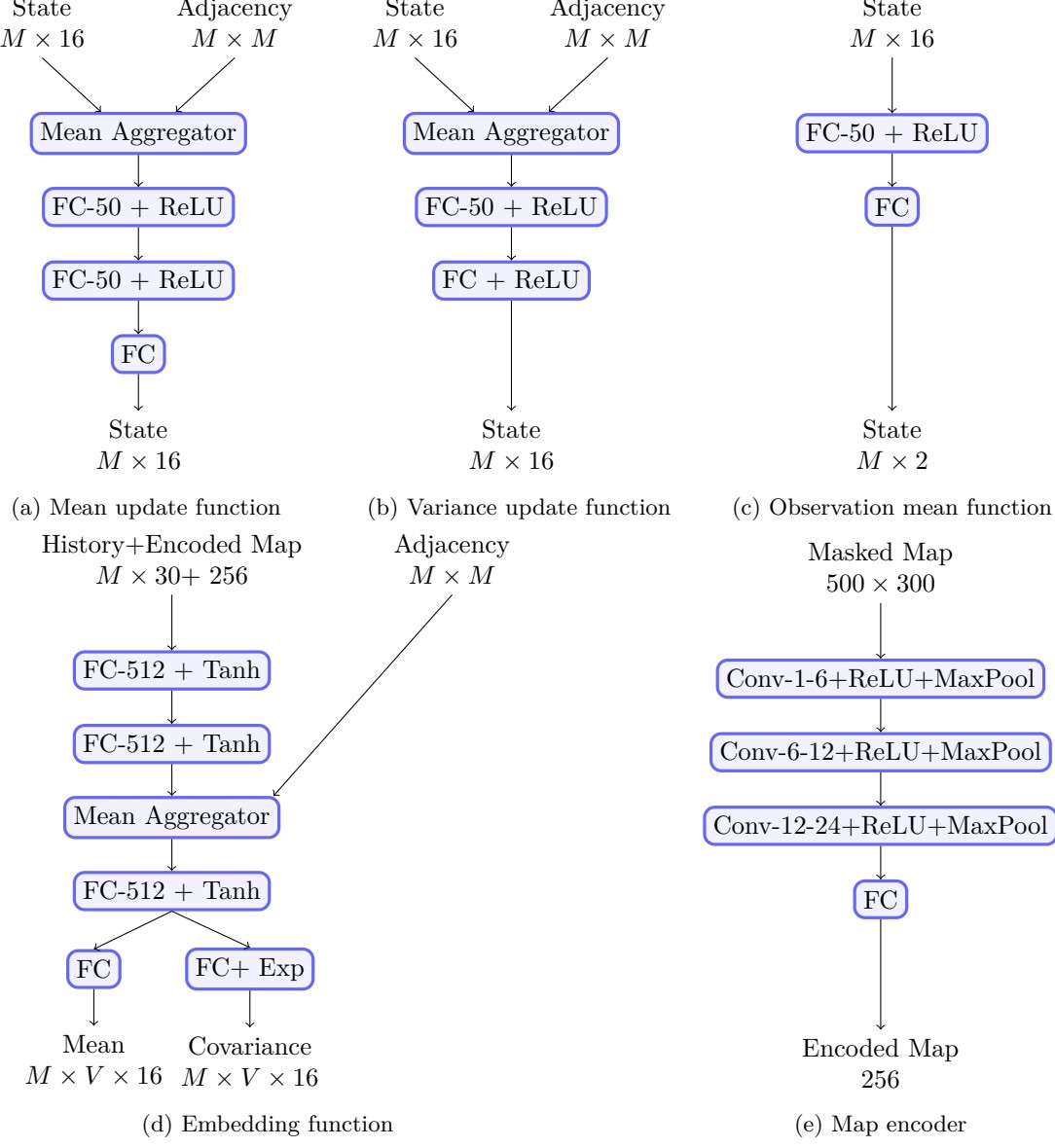

(a) Mean update function    (b) Variance update function    (c) Observation mean function

(d) Embedding function    (e) Map encoder

Figure 9: Architectures with map information. For fully connected layers we give the number of output neurons. For convolutional layers we give the number of input channels and output channels. We use a kernel size of 3 and stride 2. For MaxPool layers we use a kernel size of 2 and stride 2.

