# OpenReview forum: "Cheap and Deterministic Inference for Deep State-Space Models of Interacting Dynamical Systems"
_TMLR — Accepted by TMLR_

### Review · Reviewer_vpQ1 · 2022-09-20

**Summary Of Contributions:**

This paper introduces a state space model where the transition dynamics are parameterized by a graph neural network. This is particularly relevant in multi-agent systems, where such a model is capable of learning complex interactions between agents. An inference algorithm is proposed, which adapts an assumed density approach known as bidimensional moment matching to the particular structure of the model, which requires propagating moments through a series of nonlinearities followed by an aggregation step.

The proposed model is evaluated on a dataset of vehicular traffic in a roundabout, with ablation studies (some of which reduce to previous work) successfully demonstrating the utility of the approach.

**Broader Impact Concerns:**

No concerns regarding broader impact

**Requested Changes:**

The proposed changes, which would make me happy to raise my recommendation, essentially track the three points in the "weaknesses" section above.

1. Please re-organize sections 3 and 4 to more clearly separate the model, and the paper's methodological contributions, from the background material on inference and BMM
2. Please make explicit the objective that is used for learning the GMM parameters, and how that is handled algorithmically
3. Please clarify the experimental setting, in the context of other work on this dataset



**Strengths And Weaknesses:**

Strong aspects:

* There is a good motivation for the model, and for using graph neural networks to define transition dynamics for interacting agents
* The model itself is a sensible extension of existing work, with an interesting deterministic inference approach
* The experimental results are clearly presented, and demonstrate the utility of the method

Weak aspects:

* The contributions of the paper are not as well delineated from the existing work as they could be; in particular the organization of sections 3 and 4 is not ideal. Section 3 is introducing particular modeling choices (e.g., parameterizations of mean functions and variance functions; choice of GMM as prior over initial state), and then Section 4 covers a mix of background material (e.g. in section 4.2) as well as contributions (e.g. the moment computations for graph networks in section 4.4). This could be much easier to follow if there were a clear separation between (a) existing work, (b) modeling assumptions in this paper, (c) contributions required to apply existing work / inference algorithms to this model.
* I'm afraid I do not understand the way in which the parameters of the GNNs itself are learned in this model. While much of section 4 describes the inference algorithm, there is not much information on the learning algorithm for the network parameters. Due to the approximations made in the moment propagation steps, it doesn't seem to me that the final log likelihood approximation (e.g. as computed in Alg. 1) would be appropriate for directly running gradient descent steps for updating the GNN parameters. (Please correct me if I am wrong!) I also don't see this explicitly stated. Presumably updating the GNN parameters either requires marginalizing out x, or else sampling / imputing values of x, and then taking gradient descent steps?
* I am not familiar with this dataset and would like reassurance that this is how the data has been used in previous work. While the experiments do show that the model is successful at the chosen task, it seems like has been far simplified in terms of its generalization goals: essentially, restricting to a single roundabout (so no notion of generalization to different locations), and to 8 second segments at 5hz (i.e. with 15 frames as context, predicting an additional 25 frames). Does this match the sort of predictive task undertaken in other work looking at this data?

---

> ### Author Response · Authors · 2022-12-07
> **Response  to Reviewer vpQ1**
>
> **General:** We thank the reviewer for the positive review on our work. In the following, we
> discuss the comments and requested changes.
>
> **Comment 1:** *Please re-organize sections 3 and 4 to more clearly separate the model, and the paper's methodological contributions, from the background material on inference and BMM.*
>
> **Answer 1:** As suggested, we moved parts of Sec. 4.1 and Sec. 4.2 into the Background (Sec. 2).
>
> **Comment 2:** *Please make explicit the objective that is used for learning the GMM parameters, and how that is handled algorithmically.*
>
> **Answer 2:** We have clarified our learning procedure at the end of section 4.2 including new
> pseudo-code. In brief, we can optimize the model parameters using the PLL
> as training objective (Algorithm 2). For evaluating the PLL, we rely on our
> bidimensional moment matching scheme (Algorithm 1) as a subroutine.
>
> **Comment 3:** *Please clarify the experimental setting, in the context of other work on this dataset.*
>
> **Answer 3:** **Restriction to a single roundabout** The rounD dataset [1] has been pub-
> lished in 2020 and there exists only few works that build on it in their experiments: [2], [3] focus on multi-agent predictions and study single and multiple
> roundabouts, while the work of [4] focus on continual learning and study multiple roundabouts only.
> In our ablation study (Sec. 5.1), we decided to focus on a single roundabout
> since the main contributions of our work can be studied in this simpler setting
> and we did not want to overcomplicate things.
> With that said, we strongly agree with the reviewer that the generalization be-
> havior between different traffic environments is an important research question
> and we performed a new set of experiments to study it in more detail. Our
> results show that our model is capable of generalizing across different traffic
> environments (see Sec. 5.4. for more information). We would like to thank the
> reviewer again for coming up with this question since we believe that this new
> experiment has substantially strengthened our work.
>
> **Sample frequency**  We took our experimental setup (time horizon, sampling
> frequency) over from previous work on the NGSIM dataset [5, 6, 7]. Earlier
> work on the rounD dataset [4] and on human trajectory datasets [8, 9] employ a similar time horizon (history length of 1.6 s, prediction length of 3.2 s) and a slightly lower sampling rate of 2.5 Hz.
>
> **Citations:**
>
> [1] R. Krajewski, T. Moers, J. Bock, L. Vater, and L. Eckstein, “The rounD
> Dataset: A Drone Dataset of Road User Trajectories at Roundabouts in
> Germany,” in ITSC, 2020.
>
> [2] A. Quintanar, D. Fern ́andez-Llorca, I. Parra, R. Izquierdo, and M. Sotelo,
> “Predicting vehicles trajectories in urban scenarios with transformer net-
> works and augmented information,” in 2021 IEEE Intelligent Vehicles Sym-
> posium (IV), IEEE, 2021.
>
> [3] S. Carrasco, D. F. Llorca, and M. Sotelo, “SCOUT: Socially-consistent
> and UndersTandable graph attention network for trajectory prediction of
> vehicles and VRUs,” in 2021 IEEE Intelligent Vehicles Symposium (IV),
> IEEE, 2021.
>
> [4] H. Ma, Y. Sun, J. Li, M. Tomizuka, and C. Choi, “Continual Multi-Agent
> Interaction Behavior Prediction With Conditional Generative Memory,”
> IEEE Robotics Autom. Lett., vol. 6, no. 4, 2021.
>
> [5] N. Deo and M. Trivedi, “Convolutional Social Pooling for Vehicle Trajec-
> tory Prediction,” in CVPR Workshop, 2018.
>
> [6] C. Tang and R. R. Salakhutdinov, “Multiple Futures Prediction,” in
> NeurIPS, 2019.
>
> [7] J. Mercat, T. Gilles, N. E. Zoghby, G. Sandou, D. Beauvois, and G. P. Gil,
> “Multi-Head Attention for Multi-Modal Joint Vehicle Motion Forecasting,”
> in ICRA, 2020.
>
> [8] A. Alahi, K. Goel, V. Ramanathan, A. Robicquet, L. Fei-Fei, and
> S. Savarese, “Social LSTM: Human Trajectory Prediction in Crowded
> Spaces,” in CVPR, 2016.
>
> [9] A. Gupta, J. Johnson, L. Fei-Fei, S. Savarese, and A. Alahi, “Social GAN:
> Socially Acceptable Trajectories With Generative Adversarial Networks,”
> in CVPR, 2018.

---

### Review · Reviewer_ZrYM · 2022-10-24

**Summary Of Contributions:**

This work proposes a model and inference scheme for modeling multiple interacting agents over time. The model is based on graph neural networks, which are used to explicitly encode constraints on which agents can interact. The inference scheme is based on bidirectional moment matching (BMM). The authors specialize BMM to the graph neural network setting and evaluate the model on traffic datasets, finding that their method improves over baselines.

**Requested Changes:**

The main requested changes are:

1. Clarify how your moment-matching scheme fits a mixture of Gaussians.
2. Clarify your evaluation methodology, specifically how you estimated the negative log likelihood and why you picked the best mode for computing the RMSE.
3. Clarify your baseline implementations.

These clarifications are critical in securing my recommendation for acceptance. However, clarification alone is insufficient. It is possible that clarifying these items will convince me that the author's method or evaluation are incorrect.

**Strengths And Weaknesses:**

## Strengths

* The problem is important and well-founded.
* The proposed method explores an interesting direction.

## Major Weaknesses

Unfortunately, overall the paper is unconvincing because of a lack of clarity along with questionable methods and evaluation. The major issues I see are as follows:

### “For many prediction tasks, the quantity of interest is the predictive log-likelihood”

Please provide a citation (other than Look et al.) and further explanation for this. Why would it be desirable to only match the marginals of the data? The structure in the full joint distribution is extremely important -- without it, trajectories generated by the model may lack coherent temporal structure and instead only match the marginals at each timestep.

Why is the likelihood of the initial observation $y_0$ dropped from the ‘predictive log likelihood’?

The scheduled sampling paper only supports the idea that it is important to have matching train loss and test evaluation. The text of your paper seems to suggest it supports more -- please make it clear which claim the scheduled sampling paper is used to support.

Finally, you claim that the predictive log likelihood is the quantity of interest, but then report negative log likelihood in evaluation. As defined in your paper, the negative log likelihood and predictive log likelihoods are quite different. After the emphasis on the PLL in the main text, why only report the NLL? It makes the claims in the methods section seem suspect.

### Monte Carlo baseline methods

You repeatedly claim MC-based methods are slow to converge and perform worse. You provide scant details about your MC-based baseline, however. More information about the training method used is need to determine if the baseline is correct and strong. The current state of the art methods for training nonlinear dynamical system models use sequential Monte Carlo, such as NASMC and VSMC/FIVO/AESMC.

NASMC: Gu et al. 2015, https://arxiv.org/abs/1506.03338

VSMC: Naesseth et al. 2017, https://arxiv.org/abs/1705.11140

FIVO: Maddison et al. 2017, https://arxiv.org/abs/1705.09279

AESMC: Le et al. 2017, https://arxiv.org/abs/1705.10306

### Your moment matching scheme is possibly unfounded for mixtures of Gaussians

Your model uses a mixture of Gaussians as a prior over its initial latent state, and claims to approximate nonlinear functions of this distribution (e.g. the application of neural network layers) using moment matching. This scheme is inadequately described and is possibly incorrect. It seems to be based broadly on the assumption that moment matching can be applied to mixtures of Gaussians by applying it individually to each component. Fitting a mixture of Gaussians via moment matching requires a large number of moments, not just two as used in this work. With a maximum entropy assumption (very common) the distribution that matches two moments is the Gaussian, not a mixture of Gaussians. This casts doubt on this work's approximation of each latent state using a mixture of Gaussians.

Please clarify how your moment matching scheme fits a mixture of Gaussians at each step.

### General issues with evaluation

Your evaluation metrics are unclear, and possibly incorrect. You claim to evaluate the negative log likelihood of your models, however the negative log likelihood is intractable for nonlinear latent variable models as stated in the main text of your paper. Details on your approximation scheme for the negative log likelihood should be included. A good method for this setting is SMC with a large number of particles.

Similarly, you should integrate out the latents in your model to compute the RMSE, but you are unclear on this point. You state in tables 1 and 2, “For the case of predictors with multiple modes we calculate the RMSE for each mode and report the lowest”. This methodology is incorrect, and could largely explain the gains claimed by incorporating multimodal state priors.

## Minor Weaknesses

### Incorrect and overly-broad claims.

Many statements made in the introduction are overly broad or incorrect. Please narrow them or provide citations. For example:

**“Deterministic systems, such as complex physical simulators”**

Many simulators are not deterministic and include a random component.

**“There are two established model families that can account for model uncertainty in dynamical systems"**

This is overly broad. Stochastic dynamical systems research is incredibly diverse, with over a century of history. Furthermore, the two classes a) and b) that follow are unclear and seem to overlap. It would be good to clarify these categories and narrow your claims.

**“However, propagating a distribution through a non-linear recurrent system, as it is required in this case, cannot be performed in closed-form, and existing methods build on Monte Carlo (MC) simulations.”**

Again, this is not true in general. Please narrow your claims, clarify what you mean, and provide citations.

**“​​and well-calibrated predictions compared to state-of-the-art alternatives”**

You did not check the calibration of your model, so please do not make this claim.

**"In autonomous driving, the initial latent state can be estimated from historical information and is closely linked to the drivers’ intentions.”**

This is overly broad and must be untrue in general. The content of a latent state is model-specific.

**“They come in handy for applications in which the true underlying dynamics are not known and must be estimated from data.”**

This is also true for linear state-space models.

**“All of these approaches have in common that a good approximation to the smoothing distribution, i.e. p(x0, . . . , xT |y0, . . . , yT , I), is central for the algorithm to succeed.”**

This is not true.

### Typos and grammatical errors

There are numerous typos and grammatical errors. For example, in the introduction:

**“Allow to learn”**

**“In order to account for this increasing model complexity, large network architectures are necessary that can restrict its usage on embedded systems with limited memory capacity”**

**“Our model is capable of predicting multiple modes, which we efficiently approximate as a GMM and takes interaction into account by using GNNs.”**

Please fix those and others.

---

> ### Author Response · Authors · 2022-12-07
> **Response (I/II) to Reviewer ZrYM**
>
> **General:** We thank the reviewer for the constructive and well-thought review. We agree
> that not all parts of the manuscripts were well explained. We apologize for
> this and have tried our best to correct these flaws in the revised version of our
> manuscript. In the following, we discuss the comments and requested changes
> in more detail.
>
> **Comment 1:** *“For many prediction tasks, the quantity of interest is the predictive log-likelihood”.
> Please provide a citation (other than Look et al.) and further explanation for
> this. Why would it be desirable to only match the marginals of the data? The
> structure in the full joint distribution is extremely important – without it, trajectories generated by the model may lack coherent temporal structure and instead only match the marginals at each timestep.*
>
> **Answer 1:** **Matching training and evaluation criterion** We deem the choice of training objective application dependent and added the following explanation to Sec.
> 4.1. of the manuscript: ”The PLL is most useful for tasks that can be solved by
> assessing the marginal distributions only. An important and large application
> class that falls into this category is in the context of autonomous driving in
> which the marginal distributions of neighboring traffic participants at a given
> time horizon are often sufficient in order to control the car (e.g. [1]). As a consequence, many papers in the autonomous driving literature report as evaluation
> metrics the performance of their method at fixed time intervals which matches
> our PLL objective (e.g. [2, 3, 4]). In contrast, optimizing the joint log-likelihood
> is preferred for tasks that require sampling realistic looking trajectories, as it is
> done for instance in sentence generation [5].”
>
> **PLL as training objective in the literature** We added references to previous works that trained dynamics models on multi-step ahead predictive log
> likelihood (see also review from CaF5) to Sec. 4.1
>
> **Comment 2:** *Why is the likelihood of the initial observation dropped from the ‘predictive
> log likelihood’?*
>
> **Answer 2:** The indexing of the observations was inconsistent in the manuscript. We fixed
> this issue in the revised version. The initial latent state $x_0$ does not emit an
> observation $y_0$.
>
> **Comment 3:** *The scheduled sampling paper only supports the idea that it is important to have
> matching train loss and test evaluation. The text of your paper seems to suggest
> it supports more – please make it clear which claim the scheduled sampling paper
> is used to support.*
>
> **Answer 3:** We updated the sentence to: ”The observation that using one-step ahead predictions in training is not sufficient in order to obtain reliable multi-step ahead predictions during testing has also been made in [6] where the authors propose a scheduled sampling strategy in order to gradually switch from single to
> multi-step ahead predictions during training.”
>
> **Comment 4:** *You claim that the predictive log likelihood is the quantity of interest, but then
> report negative log likelihood in evaluation. As defined in your paper, the negative
> log likelihood and predictive log likelihoods are quite different. After the emphasis
> on the PLL in the main text, why only report the NLL? It makes the claims in
> the methods section seem suspect.*
>
> **Answer 4:** We apologize for being unclear and have clarified our evaluation metrics in App.
> D. In a nutshell, we use the predictive negative log-likelihood (NLL)
>
> \begin{equation}
> \text{NLL}(t) = -\frac{1}{N}
> \sum_{n}^N
> \sum_m^{M(n)} \frac{\log p(y_{t,n}^m|\mathcal{I})}{M(n)}
> \end{equation}
>
> that aligns well with our training objective and also follows the standard evaluation metric in the literature.
>
> **Citations:**
>
> [1] M. Herman, J. Wagner, V. Prabhakaran, N. M ̈oser, H. Ziesche, W. Ahmed,
> L. B ̈urkle, E. Kloppenburg, and C. Gl ̈aser, “Pedestrian Behavior Prediction
> for Automated Driving: Requirements, Metrics, and Relevant Features,”
> IEEE Transactions on Intelligent Transportation Systems, vol. 23, no. 9,
> 2022.
>
> [2] N. Djuric, V. Radosavljevic, H. Cui, T. Nguyen, F.-C. Chou, T.-H. Lin,
> N. Singh, and J. Schneider, “Uncertainty-aware Short-term Motion Predic-
> tion of Traffic Actors for Autonomous Driving,” in IEEE WACV, 2020.
> [3] A. Jain, S. Casas, R. Liao, Y. Xiong, S. Feng, S. Segal, and R. Urtasun,
> “Discrete Residual Flow for Probabilistic Pedestrian Behavior Prediction,”
> in CoRL, 2019.
>
> [4] Y. Chai, B. Sapp, M. Bansal, and D. Anguelov, “MultiPath: Multiple Prob-
> abilistic Anchor Trajectory Hypotheses for Behavior Prediction,” arXiv,
> vol. abs/1910.05449, 2019.
>
> [5] A. Vaswani, N. Shazeer, N. Parmar, J. Uszkoreit, L. Jones, A. N. Gomez,
> L. Kaiser, and I. Polosukhin, “Attention Is All You Need,” in NeurIPS,
> 2017.
>
> [6] S. Bengio, O. Vinyals, N. Jaitly, and N. Shazeer, “Scheduled Sampling for
> Sequence Prediction with Recurrent Neural Networks,” in NeurIPS, 2015.

---

> ### Author Response · Authors · 2022-12-07
> **Response (II/II) to Reviewer ZrYM**
>
> **Comment 5:** *You repeatedly claim MC-based methods are slow to converge and perform
> worse. You provide scant details about your MC-based baseline, however. More
> information about the training method used is need to determine if the baseline
> is correct and strong. The current state of the art methods for training nonlinear dynamical system models use sequential Monte Carlo, such as NASMC and VSMC/FIVO/AESMC.*
>
> **Answer 5:** We provide more details about the existing MC baseline in Sec. 5.1. In brief,
> we used the same routines as for our method with the only difference that we
> used Monte Carlo sampling for propagating the latent state through the neural
> network layers. In addition, we included two new comparison partners to our benchmark in Sec.
> 5.1. using variational inference [7] and sequential Monte Carlo [8] for training.
> Our new experiments confirm the benefits of our method.
>
> **Comment 6:** *Your moment matching scheme is possibly unfounded for mixtures of Gaussians
> Your model uses a mixture of Gaussians as a prior over its initial latent state,
> and claims to approximate nonlinear functions of this distribution (e.g. the
> application of neural network layers) using moment matching. This scheme is
> inadequately described and is possibly incorrect. It seems to be based broadly on
> the assumption that moment matching can be applied to mixtures of Gaussians
> by applying it individually to each component. Fitting a mixture of Gaussians via
> moment matching requires a large number of moments, not just two as used in
> this work. With a maximum entropy assumption (very common) the distribution
> that matches two moments is the Gaussian, not a mixture of Gaussians. This
> casts doubt on this work’s approximation of each latent state using a mixture of
> Gaussians. Please clarify how your moment matching scheme fits a mixture of
> Gaussians at each step.*
>
> **Answer 6:** We provide clarification about our moment matching scheme in Sec. 4.2 in
> which we (i) prove that our scheme is exact for locally linear dynamical systems and (ii) illustrate its behavior on a toy example for which our method is able to recover the ground-truth dynamics.
>
> **Comment 7:** *Your evaluation metrics are unclear, and possibly incorrect. You claim to
> evaluate the negative log likelihood of your models, however the negative log
> likelihood is intractable for nonlinear latent variable models as stated in the
> main text of your paper. Details on your approximation scheme for the negative
> log likelihood should be included. A good method for this setting is SMC with a
> large number of particles. Similarly, you should integrate out the latents in your
> model to compute the RMSE, but you are unclear on this point. You state in
> tables 1 and 2, “For the case of predictors with multiple modes we calculate the
> RMSE for each mode and report the lowest”. This methodology is incorrect, and
> could largely explain the gains claimed by incorporating multimodal state priors.*
>
> **Answer 7:** **RMSE** We agree that the calculation of the RMSE is rather untypical out-
> side of the traffic forecasting domain. However, our evaluation scheme follows
> concurrent literature, which we support by several citations. This evaluation
> procedure is also commonly used for public traffic forecasting challenges (e.g.
> https://eval.ai/web/challenges/challenge-page/1719/evaluation).
> **NLL** Please see our answer above.
>
> **Minor Comments+Answers:**
>
> **Incorrect and overly broad claims:** As suggested, we refined our claims and
> provide citations whenever possible in the updated version of the manuscript.
>
> **Typos and grammatical errors:**  We thank the reviewer for pointing out these
> errors and have corrected them in the new version.
>
> **Citations:**
>
> [7] R. G. Krishnan, U. Shalit, and D. Sontag, “Structured Inference Networks
> for Nonlinear State Space Models,” in AAAI, 2017.
>
> [8] C. J. Maddison, J. Lawson, G. Tucker, N. Heess, M. Norouzi, A. Mnih,
> A. Doucet, and Y. Teh, “Filtering Variational Objectives,” in NeurIPS,
> 2017.

---

> > ### Comment · Reviewer_ZrYM · 2022-12-29
> > **Still unconvinced re: reporting the result from the best mode**
> >
> > Thank you for your updates to your paper, the revised version is stronger. There is still one remaining crucial issue that I do not fully understand: Why do you report the RMSE from the best mode for the multi-modal models? That does not seem to accurately represent test-time performance where the best mode will be unknown. Please provide justification for this approach or change to integrating out the choice of mode. If this can be resolved I will vote for accepting the paper.

---

> > > ### Author Response · Authors · 2023-01-09
> > > **Response to Reviewer ZrYM**
> > >
> > > We thank the reviewer for the positive feedback on the revised version.
> > >
> > > The minRMSE value assesses if the model produces a set of diverse predictions
> > > where at least one proposal trajectory is close to the true outcome. While this
> > > score gives us useful information about the model capacity, we agree that the
> > > minRMSE from the best mode does not give a complete representation of the
> > > test-time performance.
> > > We have therefore decided to move the minRMSE values into App. F and added
> > > RMSE values, where we now integrate out the choice of mode before making
> > > final predictions, in the main text (Table 1 and 2). Our results show that
> > > our approach compares favorably against state-of-the-art alternatives. When
> > > increasing the number of modes in our model, we observe that the RMSE is
> > > not affected, while the NLL is decreased significantly. This behavior can be
> > > attributed by the different properties of the scores: RMSE takes only information from the first moment (expectation value) into account, whereas the
> > > NLL is a strictly proper scoring rule[1] that incorporates information from all
> > > moments.
> > > We explain the difference between the RMSE variants, as well as their pro and
> > > cons, in more detail in App. D.
> > >
> > > [1] T. Gneiting and A. E. Raftery, “Strictly proper scoring rules, prediction,
> > > and estimation,” Journal of the American statistical Association, vol. 102, no. 477, 2007.

---

### Review · Reviewer_CaF5 · 2022-11-23

**Summary Of Contributions:**

The authors introduce Graph Deep State Space Models (GDSSM) to learn state space models over $M$ interacting agents. The contributions of this work are twofold: 1) the combination of classical deep-state space models with graph neural networks and 2) a deterministic training scheme based on moment matching, combined with a multi-modal prior, allows for efficient training of the model while allowing for expressivity.

**Broader Impact Concerns:**

There are no concerns on the ethical implications.

**Requested Changes:**

1. Eq. 14 is only true if you model the transition dynamics is $x_t = x_{t-1} + f(x_{t-1})$ e.g, $f$ is learning the velocity. As a simple counter-example, if $x_t = Ax_{t-1}$ and $x_t \sim \mathcal{N}(\mu, \Sigma)$ then $\mathbb{E}[x_t] = A\mu$ while if $x_t = x_t + Ax_{t-1}$ then $\mathbb{E}[x_t] = \mu + A\mu$. Please update the previous state space model equations to reflect this.

2. For completeness, it would be nice if the authors reviewed how to propagate means and covariances through the common nonlinearities, or at least for the non-linearities used in the paper.

3. Some key references are missing. Mainly, there have been previous works on training dynamics models on multi-step ahead predictive log likelihood: https://arxiv.org/abs/1811.04551 and https://arxiv.org/abs/2106.06064.

4. Training details need to be added to make this work reproducible.

5. In section 5.2.1, the authors state: *"There exists a large body of prior work [...] which applies GNNs for traffic forecasting on the NGSIM
dataset and provide quantification of predictive uncertainty" but none of the methods compared against use GNNs! For completeness, it would be nice if the authors also compared against a previously proposed method or, at the very at least, include GNN variants of the methods they did compare against, e.g., the output of the GNN can be fed into an LSTM as input.

**Strengths And Weaknesses:**

# Strengths

I love the simplicity and motivation behind the method! Moreover, I think the writing is top-notch and does a fantastic job describing the many working pieces of the proposed method. Moreover, the ablation method is **very** extensive and demonstrates not only the importance of individual pieces (non-linear vs linear dynamics, GNN vs no agent interaction, deterministic vs MC) but also provides an extensive evaluation of several different approximations for the covariance.

# Weaknesses

In this section, I will discuss weaknesses, questions, and comments. Firstly, it seems at each time instant, only one round of message passing is done, while to my understanding, multiple rounds of message passing can be done at each time point. Is there a reason this was done? It would be nice if this were discussed.

Next, on page 5 the authors state: *"Furthermore, we note that although the transition noise factorizes across agents, our model is capable of modeling correlations between agents since the mean and the variance depend not only on the state of the m-th agent, but also on the states of all neighboring agents. In consequence, after $a$ aggregation steps, our model accounts for correlations between agent m and agent m′ provided that they are connected by a path that is at most a steps long."*. This isn't unique to the proposed approach but rather is a property of using transition functions for the mean and variance: for instance, if I used
$$ p(x_t \mid x_{t-1}) = \mathcal{N}(x_t \mid Ax_{t-1}, \textrm{diag}\left(\exp( Bx_{t-1})) \right) $$
then this would allow correlations between agents to propagate.

Lastly, there are no training details whatsoever for any of the experiments, making the paper nearly impossible to reproduce. Neither in the main paper or the appendix, do I see any details on batch size, learning rate, optimizer, etc.

---

> ### Author Response · Authors · 2022-12-07
> **Response to Reviewer CaF5**
>
> **General:** We thank the reviewer for the positive comments on (i) our method, (ii) the
> experiments, and (iii) the writing. In the following, we discuss the comments
> and requested changes.
>
> **Comment 1:** *It seems at each time instant, only one round of message passing is done,
> while to my understanding, multiple rounds of message passing can be done at
> each time point. Is there a reason this was done? It would be nice if this were
> discussed.*
>
> **Answer 1:** Yes, there can be done multiple rounds of message passing. We have clarified
> our choice of doing a single round of message passing in Sec. 3: ”Note that
> our transition model consists of a single aggregation and update step which is
> sufficient if the data is densely sampled such that the information flow between
> agents is fast compared to the evolution of the state dynamics. However, it is
> also straight-forward to extend the model to multiple message-passing steps per
> time point by stacking multiple GNN layers in the mean and variance update
> function.”
>
> **Comment 2:** *”Furthermore, we note that although the transition noise factorizes across
> agents, our model is capable of modeling correlations between agents since the
> mean and the variance depend not only on the state of the m-th agent, but also
> on the states of all neighboring agents. In consequence, after aggregation steps,
> our model accounts for correlations between agent m and agent m provided that
> they are connected by a path that is at most a steps long.”. This isn’t unique to
> the proposed approach but rather is a property of using transition functions for
> the mean and variance.*
>
> **Answer 2:** We agree with the comment and clarified the statement to: ”Furthermore, we
> note that although the transition noise factorizes across agents, correlations
> between agents emerge since the mean and the variance depend not only on the
> state of the m-th agent, but also on the states of all neighboring agents. After
> a aggregation steps, our GNN model accounts for correlations between agent
> m and agent m′ provided that they are connected by a path that is at most a
> steps long. In contrast, methods, that only take the state of the m-th agent into
> account, do not lead to any correlations, while methods, that take the state of
> all other agents into account, lead to a fully correlated covariance matrix after
> one time step.”
>
> **Comment 3:** *Eq. 14 is only true if you model the transition dynamics is e.g, is learning the
> velocity. As a simple counter-example, if and then while if then . Please update
> the previous state space model equations to reflect this.*
>
> **Answer 3:** We have clarified that Eq. 14 (now Eq. 6) holds only if the one-step transition
> kernel follows the transition model in Eq. 2
>
> **Comment 4:** *For completeness, it would be nice if the authors reviewed how to propagate
> means and covariances through the common nonlinearities, or at least for the
> non-linearities used in the paper.*
>
> **Answer 4:** We added moment propagation rules for the ReLU activation to App. B.
>
> **Comment 5:** *Some key references are missing. Mainly, there have been previous works on
> training dynamics models on multi-step ahead predictive log likelihood*
>
> **Answer 5:** Thanks for pointing us to these interesting works! We added the citations to
> Sec. 4.1.
>
> **Comment 6:** *Training details need to be added to make this work reproducible.*
>
> **Answer 6:** We agree about the importance of making our work reproducible. Therefore,
> (i) we added training details for our experiments to App. C and (ii) will make
> the code publicly available after acceptance.
>
> **Comment 7:** *In section 5.2.1, the authors state: ”There exists a large body of prior work
> [...] which applies GNNs for traffic forecasting on the NGSIM dataset and pro-
> vide quantification of predictive uncertainty” but none of the methods compared
> against use GNNs! For completeness, it would be nice if the authors also com-
> pared against a previously proposed method or, at the very at least, include GNN
> variants of the methods they did compare against, e.g., the output of the GNN
> can be fed into an LSTM as input.*
>
> **Answer 7:** We added the method ST-LSTM [1], which combines a GNN with a deterministic LSTM, to our comparison on the NGSIM dataset in Sec 5.3.
>
> **Citations:**
>
> [1] G. Chen, L. Hu, Q. Zhang, Z. Ren, X. Gao, and J. Cheng, “ST-LSTM:
> Spatio-Temporal Graph Based Long Short-Term Memory Network For Ve-
> hicle Trajectory Prediction,” in IEEE ICIP, 2020.

---

### Author Response · Authors · 2022-12-07
**Response to all Reviewers**

We thank the reviewers for constructive and well-thought reviews. We hope to
have appropriately addressed their comments.

We provide individual feedback to the reviewer’s comments below and adjusted
the manuscript accordingly, taking also all minor remarks into account. For
easy readability we have colour-coded the changes in the manuscript.

---

### Author Response · Authors · 2023-03-01
**Camera ready version**

Dear AE and reviewers,

Thank you to all for the constructive feedback during the rebuttal period.

We have uploaded the camera ready version. In addition, we provide the link to the code of our paper.  The repository is currently empty and we will upload our code soon.

---

### Decision · Action_Editors · 2023-02-22

**Recommendation:** Accept as is

**Comment:**

This article addresses an interesting topic, how to model interacting stochastic systems and how to make inference for such systems. The paper has now been reviewed by three reviewers: two recommend accept (ZrYM  and CaF5) and one leans towards acceptance (vpQ1). The authors have updated substantially the paper at the revision stage and all the reviewers are now happy with the re-organized manuscript and the writing. They agree that the problem addressed is important and that the proposed methodology is principled and explores an interesting direction. Initial questions about some of the technical aspects of the methodology (moment matching) have been addressed satisfactorily by the authors and all reviewers do believe the work is technically sound and well evaluated.

I thus believe the material should be published as it is. The authors should make their code accessible for reproducibilty as promised in their response to reviewers.

**Audience:**

There is a lot of interest in graph neural networks and time series so it is clear that it will be of interest to many TMLR readers.

**Claims And Evidence:**

The authors introduce a novel model for learning state space models over interacting agents, which they call Graph Deep State Space Models (GDSSM). GDSSM combines ``classical" deep-state space models with graph neural networks to explicitly encode constraints on agent interactions. The model's transition dynamics are parameterized by a graph neural network, which allows it to learn complex interactions between agents in multi-agent systems. The inference scheme is based on a bidirectional moment matching (BMM) algorithm, specialized to the graph neural network setting, which efficiently propagates moments through nonlinearities followed by an aggregation step. The proposed model is evaluated on a dataset of vehicular traffic in a roundabout, with ablation studies showing that the approach improves over baselines. Overall, the contributions of this work are twofold: the combination of deep-state space models with graph neural networks and the deterministic training scheme based on moment matching, which allows for efficient training while retaining expressivity. The proposed method is principled and the claims of the authors are substantiated.